# Frog Density and Growth Stage of Rice Impact Paddy Field and Gut Microbial Communities in Rice–Frog Co-Cropping Models

**DOI:** 10.3390/microorganisms13071700

**Published:** 2025-07-20

**Authors:** Zhangyan Zhu, Ran Li, Yunshuang Ma, Anran Yu, Rongquan Zheng

**Affiliations:** 1Provincial Key Laboratory of Wildlife Biotechnology and Conservation and Utilization, Zhejiang Normal University, Jinhua 321004, China; jjzhuzhangyan@163.com (Z.Z.); mys000408@zjnu.edu.cn (Y.M.); yuanran@zjnu.edu.cn (A.Y.); 2Xingzhi College, Zhejiang Normal University, Jinhua 321004, China; ytliran@126.com

**Keywords:** rice–frog co-cropping mode, black-spotted frog (*Pelophylax nigromaculatus*), water microbiota, gut microbiota

## Abstract

The black-spotted frog (*Pelophylax nigromaculatus*) is a common economic species in the rice–frog ecological cropping mode. The present study investigated microbial community structures in paddy water and black-spotted frog’s guts across rice monoculture and low-/high-density rice–frog co-cropping systems at four rice growth stages. Proteobacteria dominate in paddy water, while frog guts are enriched in Firmicutes and Actinobacteriota. The frog density shows no impact on the α-diversity, but rice growth stages significantly alter the Shannon, Simpson, and Pielou_e indices (*p* < 0.01). Co-cropping may promote amino acid synthesis, elemental cycling, and stress tolerance in paddy water microbiota, which are more diverse than gut microbiota. Strong correlations exist between paddy water and gut microbiotas, with *Limnohabitans* being linked to gut diversity (*p* < 0.05). Low-density co-cropping enhances *Xenorhabdus*, which is beneficial for pest control and stabilizes gut microbiota. The results of this study offer insights for managing rice–frog systems based on rice growth stages.

## 1. Introduction

With the increasing global human population and growing urbanization, the agricultural sector faces many global challenges, such as climate change and environmental pollution caused by the misuse of pesticides and fertilizers, which will continue to intensify in the face of food shortages in the future [1,2,3]. At present, the global production and consumption of rice (*Oryza sativa* L.) are increasing [4,5]. However, due to the turbulent international situation and the poor efficiency of rice monoculture, farmers’ motivation to plant rice has been negatively affected. Furthermore, the phenomenon of the abandonment of low- and medium-yielding paddy fields is more serious in some areas, and has resulted in worldwide food supply shortages and global food security concerns [6,7,8]. Within this context, China’s agricultural and rural situation is undergoing profound changes, with the large-scale conversion of agricultural lands to non-agricultural lands such as land for housing, industry, and transportation [9]. Thus, the area of aquaculture is decreasing year by year, which seriously restricts the sustainable development of aquaculture [10].

To improve the utilization of land and water resources, rice–frog co-cropping has become an emerging model for rice-fishery co-cropping. This mode can leverage the symbiotic and mutually beneficial effects of rice and frogs, achieving a double harvest of organic rice and organic frogs [11]. The rice–frog co-cropping model enables farmers to obtain economic benefits from frog farming, which improves their income compared with the single cultivation of rice and thus stabilizes their incentive to grow food [12,13]. In terms of ecological farming and environmental protection, in the rice-fishery co-cropping model, the presence of natural enemies of pests reduces the dependence on pesticides and chemical fertilizers by manure fertilizing the fields, effectively reducing the non-point source (NPS) pollution from agriculture and improving the quality of rice [14]. Furthermore, Ma et al. [15] found that the activities of frogs improved the texture and nutrient status of soil, which promoted the growth and development of rice and thus increased the rice yield. In addition, the introduction of frogs into rice fields can effectively increase the richness and diversity of the bacterial communities in soil and improve the efficiency of soil nitrogen utilization [16]. Rapidly increasing the economic yield of reclaimed cropland while reducing the environmental impacts of chemical fertilizers and improving the water and soil biological conditions of paddy fields has become a key issue in ecological research on the rice-fishery cropping model [17].

Frogs are an important economic source for the rice–frog co-cropping model. Black-spotted frogs (*Pelophylax nigromaculatus*) are one of the most widely distributed amphibians in southern China and are also found in Japan, South Korea, North Korea, and Russia. They not only have tender meat, an excellent flavor, and rich nutrition (high in protein and low in fat), but also have medicinal value due to reducing swelling, aiding detoxification, and relieving cough [18]. Thus, they are highly favored by farmers, and in recent years, the culture of the black-spotted frog has already increased, which has made it one of the characteristic species that is used under the ecological rice–frog co-cropping mode [18].

The gut microbiota plays an important role in the growth and development (including immune response, digestion, metabolism, etc.) of the host [19,20,21,22,23]. Amphibian gut bacteria are mainly derived from the surrounding water environment through their feeding, and the structure and diversity of their gut microbiota constantly change with the growth and development of the host and changes in the surrounding water environment [24,25]. Thus, the diet and living environment of amphibian species can influence the structure and diversity of their gut microbiota [26,27]. Previous work on the relationship between black-spotted frogs and soil microbiota showed that rice–frog co-cropping models can modify environmental factors, such as increasing soil fertility and microbial diversity in reclaimed land [15]. Similarly, paddy water is one of the most important surrounding environments for black-spotted frogs in the rice–frog co-cropping models [28,29], but the relationship between the paddy water microbiota and black-spotted frog gut microbiota has not been studied. In addition, changes in frog density may have an impact on the microbial diversity by affecting the frequency of contact between individuals, competitive pressures, and so on, which could in turn affect the health and ultimately the microbial diversity of the frogs. However, there are few reports on the effects of density on the microbial structure of frogs’ guts.

In this study, the relationships between paddy water and the gut microbiota of black-spotted frogs under a rice monoculture and different densities in rice–frog co-cropping models were investigated. Additionally, the effects of rice–frog co-cropping on the gut microbiota of frogs were investigated to provide more scientific references for rice–frog co-cropping and the ecological breeding of the black-spotted frog.

## 2. Materials and Methods

### 2.1. Study Area Overview

The study site was located in Shafanxiang (119.233~120.775° N, 28.533~29.683° E), Jinhua City, Zhejiang Province, China. It has hilly topography and a subtropical monsoon climate characterized by simultaneous rain and heat in summer, and mild and low rainfall in winter, with an annual rainfall of 1309 mm and an annual average of 1810 h of sunshine [15]. The experiment was carried out in 2022 from July to September.

### 2.2. Sample Collection and Processing

In this study, “Yong You No. 31” rice and black-spotted frogs were selected as the rice and frog species, respectively, for the rice–frog co-cropping model. The frogs selected were healthy, disease-free, and weighed 1~5 g. To ensure treatment homogeneity, frogs were visually sorted by size (snout-vent length: 1.6~1.9 cm) and mass (1.5~2.0 g) at the start of the experiment. A pre-experiment *t*-test confirmed no significant mass difference between low- (1.733 ± 0.218 g) and high- (1.654 ± 0.265 g) density groups (n = 30, t = 1.254, *p* = 0.215). All frogs were immersed in 2%~3% salt water for 5~10 min before stocking, and the same treatment was applied to all fields, with no pesticide application during the entire period. A previous study on rice–frog co-cropping has confirmed that the density of frogs required to achieve optimal ecological and economic benefits is 60,000 frogs/ha (6 frogs/m^2^) [30]. Since the weight of the tiger frog (*Rana rugulosa* Wiegmann, 1834) used in the cited study (15 g on average) was much higher than that of the black-spotted frog (*P. nigromaculatus* Hallowell, 1861) used in the present experiment (1–5 g), the density of frogs was increased in this experiment.

Three experimental production fields were selected and divided into a rice monoculture group (0 frogs/mu, WM represents the water in this group), a low-density rice–frog co-cropping group (5000 frogs/mu, corresponding to 8 frogs/m^2^, WL and IL separately represent the water and gut of frogs in this group), and a high-density rice–frog co-cropping group (10,000 frogs/mu, corresponding to 15 frogs/m^2^, WH and IH separately represent the water and gut of frogs in this group). Fifteen days after transplanting the rice, the seedlings had become green and viable, and the prepared frogs were placed in the three rice fields on a sunny day. Jindadi feed (Zhejiang Jindadi Bio-technology Co., Ltd., Shaoxing, China) was used as the feed for the frogs in this study. The feeding amount was approximately 5% of the frog’s weight. The frog feed contained ≥40% crude protein, ≥4.5% crude fat, ≤8% crude fiber, ≤18% crude ash, ≥1% total phosphorus, and ≥1.8% lysine. After being put in the paddy field, the frogs were fed at 17:00 every day.

Water samples were collected from the rice monoculture group, the low-density rice–frog co-cropping group, and the high-density rice–frog co-cropping group at four periods (the tillering stage, the embryonic stage, the full heading stage, and the ripening stage) of rice development, with a sampling depth of 10 cm and three replicates per group. After collection, the water samples were immediately filtered through a 200-mesh sterilized sieve. Subsequently, a high-quality 0.22 μm filter membrane (Millipore) was used to filter the samples to effectively extract the microbiota. The extracted membranes were then placed in sterilized centrifuge tubes where they were temporarily stored in liquid nitrogen, taken back to the laboratory, and immediately frozen in an ultra-low-temperature refrigerator at −80 °C for storage.

Gut microbiota were collected from black-spotted frogs at the same time as the water samples. The intestines of five healthy black-spotted frogs of similar size were collected from each of the rice–frog co-cropping paddies at each period, with three biological replicates per group. Frogs were dissected under sterile conditions, and their guts were immediately collected in sterilized EP tubes and stored at −80 °C for DNA extraction.

### 2.3. DNA Extraction and High-Throughput Sequencing Analysis

The TGuide S96 Magnetic Bead kit for soil/fecal genomic DNA extraction (Tiangen Biochemical Science and Technology (Beijing) Co., Ltd., Beijing, China, DP812) was used to extract nucleic acids from the water and gut samples. The extracted nucleic acids were then used as templates for PCR amplification of the V3–V4 region of microbial 16S rRNA using the universal primers 338F and 806R (Table 1). The PCR was performed with a 10 µL reaction system under the following conditions: 98 °C for 30 s of denaturation, 27 cycles at 98 °C for 15 s, 50 °C for 30 s, 72 °C for 5 min of annealing, and 72 °C for 5 min for the final elongation. Then, VAHTSTM DNA Clean Beads were used to purify the amplified PCR products. After library construction and testing, the eligible sample libraries were sent to an Illumina NovaSeq 6000 sequencing platform (Illumina, San Diego, CA, USA) at Qingdao Kechuang Quality Inspection Company for high-throughput sequencing.

### 2.4. Sequencing Analysis and Microbial Taxonomic Identification

The microbiome was analyzed using QIIME2 (version 2021.8) [31]. Raw sequence data were decoded using the demux plugin, and primers were excised using the cutadapt plugin [32]. The sequences were processed for quality filtering, denoising, splicing, and chimera removal using the DADA2 plugin [33]. Sequences were then merged at 100% sequence similarity to generate characteristic sequence amplicon sequence variation (ASV) and abundance data tables. The ASV feature sequences were compared with reference sequences in the Greengenes database to obtain taxonomic information corresponding to each ASV. ASVs with abundance values below 0.001% of the total number of sequenced samples were excluded, and then the abundance matrix of the excluded rare ASVs was analyzed.

### 2.5. Data Analysis

The ASV data were analyzed using QIIME 2 (version 2021.8) to calculate the alpha-diversity (α-diversity, including the Observed_otus, Shannon, Simpson, Chao1, and Pielou_e indices) metrics and to perform non-metric multidimensional scaling analysis (NMDS) of the beta-diversity (β-diversity) in the samples, to explore changes in the microbial composition of the paddy water and the frog’s guts. With the use of SPSS 21.0 (IBM, Inc., Armonk, NY, USA), differences in the α-diversity indices of the samples from different experimental groups were tested using a one-way analysis of variance (ANOVA). Significant differences between periods and frog densities were also compared. The applicability of the data to one-way ANOVA was tested by using the Shapiro–Wilk test (*p* > 0.05). Multiple comparisons were performed using Tukey’s test or Dunnett’s T3 test depending on whether they passed Levene’s test for homogeneity.

The Silva database (Release 138) was used as a reference to annotate the feature sequences with precise taxonomy using a simple Bayesian classifier [34]. With the use of R 4.4.2, LEfSe was used to analyze the differences in the relative abundance of the microbiota of the paddy water and the guts of frogs under different culture densities [35]. Taxa with significant *p*-values (*p* < 0.05) and LDA scores ≥ 3 were considered to be differentially abundant taxa [36]. Using the Wilcoxon test and correcting for FDR by using the Benjamini–Hochberg (BH) procedure. The functional abundance of the samples was predicted in PICRUSt2 based on marker gene sequence abundance.

Microbial similarity analyses were performed based on 100% similarity ASVs, and UpSet analyses were conducted in R 4.4.2 (UpSetR) to determine exclusive and similar ASVs of the samples. Spearman analysis was used to determine the correlation between the paddy water and frog gut microbial communities at the genus level, and a correlation heatmap was plotted using R 4.4.2 (corrplot).

Based on the top 15 genera from the water microbiota and frog gut microbiota, Pearson’s correlation analysis was performed to assess the statistical significance of the differences between the water microbiota and frog gut microbiota. The relationships between the microbial diversity of paddy field water and the dominant microbe species with a mean relative abundance >1% in the guts of frogs in both low- and high-density rice co-cropping groups were analyzed by using Cramer’s V. Mantel test analysis (with 9999 permutations) [37]. Correlation coefficient r- and *p*-values were obtained from the Pearson’s correlation analysis of the dominant species in the paddy water and frog samples using the Mantel function. After that, the Mantel test was completed in the vegan and ggcor packages and then visualized and plotted.

## 3. Results

### 3.1. Paddy Field Water Microbial Community Composition

#### 3.1.1. Species Composition

After analyzing the paddy field water samples, 78 phyla, 213 classes, 496 orders, 848 families, 2078 genera, and 3541 microbiota species were detected. At the phylum level, the microbial community compositions of the water samples were similar in all three culturing modes and at all four rice stages, with the Proteobacteria being the dominant taxon of those with relative abundances above 33% (Figure 1a). In addition, the abundances of the Bacteroidota, Actinobacteriota, and Firmicutes were also relatively high, with the abundance of the Acidobacteriota increasing significantly at the ripening stage of the rice. At the genus level, the dominant microbes in the water were *Cetobacterium*, *Limnohabitans*, and *Acinetobacter* (Figure 1b), with Chloroplast_unclassified being more abundant in the rice monoculture group than in the rice–frog co-cropping groups. However, *Acinetobacter* was significantly enriched in the rice–frog co-cropping groups except during the ripening stage of the rice.

#### 3.1.2. Microbial Diversity

The α-diversity indices of the water microbiota were not significantly (*p >* 0.05) influenced by the frog culturing densities (Table 2, Figure 2). However, the observed_otus (F_3,24_ = 16.434, *p* < 0.001) and Chao1 (F_3,24_ = 16.391, *p* < 0.001) indices of the rice monoculture group (WM) and the high-density rice–frog co-cropping group (WH) were significantly affected by the different rice growth stages. Furthermore, the Shannon (F_3,8_ = 15.924, *p* < 0.001), Simpson (F_3,8_ = 6.152, *p* = 0.018), and Pielou_e (F_3,8_ = 12.105, *p* = 0.002) indices of the high-density rice–frog co-cropping group were also considerably affected by the different growth stages of the rice. In the low-density rice–frog co-cropping group, no indices were significantly impacted by the different growth stages of the rice (*p* > 0.05 for all). Overall, it appeared that the α-diversity of the water microbiota tended to increase with the growth of the rice, with the lowest diversity being seen at the tillering and embryonic stages and the highest diversity at the ripening stage.

The results of the NMDS analysis of the β-diversity of the water microbiota showed good model fitting (Stress = 0.154) (Figure 3). From left to right in Figure 3, the tillering, embryonic, full heading, and ripening stages of rice are shown. In the same period, from left to right, the rice monoculture group, the low-density rice–frog co-cropping group, and the high-density rice–frog co-cropping group are shown. Rice growth stages have a significant effect on the β-diversity of the microbiota in the paddy water, while the rice–frog co-cropping model had little impact on the β-diversity.

#### 3.1.3. LEfSe Analysis of Differential Microbiotal Species in Paddy Field Water

The LEfSe analysis of differential species in the water microbiota exhibited a large number of differentially significant species in the water of the rice monoculture and the low-density rice–frog co-cropping groups (Figure 4). Among these, there were 13 and 12 differentially significant species in the water of the rice monoculture group and the low-density rice–frog co-cropping group, respectively (*p* < 0.01). The significantly enriched microbiota in the water of the rice monoculture group were Nitrosomonadaceae_unclassified, *Bosea*, Burkholderiales_unclassified, and NS9_marine_group_unclassified (*p* < 0.001 for all). However, the only significantly enriched microbiota in the water of the low-density rice–frog co-cropping group was *Xenorhabdus* (*p* < 0.001).

#### 3.1.4. Predictive Analysis of Water Microbiota

Based on the PICRUSt2 functional prediction, there were significant differences in the functional categories of KEGG pathway 3, including phenylpropanoid biosynthesis, D-Glutamine and D-Glutamate metabolism, and cellular antigen biosynthesis, as well as penicillin and cephalosporin biosynthesis, between the three treatment groups (Figure 5).

BugBase phenotypic predictions (Figure 6) showed that the paddy field water in the rice–frog co-cropping group had higher relative abundances of ‘anaerobic’, ‘contains mobile elements’, ‘facultatively anaerobic’, ‘gram positive’, ‘potentially pathogenic’, and ‘stress tolerant’ genes compared to the rice monoculture group. Firstly, the ‘Anaerobic’ and ‘stress-tolerant’ gene abundances were significantly higher in the co-cropping group during the rice tillering, embryonic, and full heading stages, but were lower at the ripening stage (except for ’stress-tolerant’ genes, which remained high). Secondly, the ‘contains mobile elements’ gene abundance was higher in the co-cropping groups at all stages except the tillering stage. Thirdly, Gram-positive bacteria were more abundant in the co-cropping groups during the tillering and ripening stages. Lastly, the high-density co-cropping groups showed elevated numbers of potentially pathogenic microbes compared to the low-density co-cropping and monoculture groups at the tillering, embryonic, and full heading stages.

### 3.2. Gut Microbiota of Frogs

#### 3.2.1. Microbial Community Composition

After analyzing the gut samples of the black-spotted frogs, 41 phyla, 124 classes, 273 orders, 473 families, 1101 genera, and 1623 species were detected.

At the phylum level, the most abundant microbes in the guts of the black-spotted frogs were Firmicutes and Proteobacteria (Figure 7a), and an increase in the Fusobacteriota was detected in the gut of the black-spotted frog at the embryonic stage of the rice. At the genus level, the microbial community composition of the guts of the black-spotted frogs differed between groups (Figure 7b). In general, the microbiota were mainly dominated by *Lactococcus* and *Cetobacterium*, and the relative abundance of *Pseudoalteromonas* and *Vibrio* was also high. These genera were more abundant in the guts of the black-spotted frogs in the high-density group at the tillering and embryonic stages of rice, but less abundant in the high-density group at the full heading and ripening stages.

#### 3.2.2. Gut Microbial Diversity

The α-diversity indices of the gut microbiota of the frogs were not significantly influenced by the culturing densities (Figure 8, Table 3). However, the Shannon (F_3,8_ = 8.454, *p* = 0.007), Simpson (F_3,8_ = 13.622, *p* = 0.002), and Pielou_e (F_3,8_ = 13.772, *p* = 0.002) indices of the low-density rice–frog co-cropping group were significantly affected by the different growth stages of the rice. Overall, in the low-density rice–frog co-cropping group, the gut microbial diversity of the black-spotted frogs was the lowest at the rice embryonic stage and relatively high at the other stages.

The results of the NMDS analysis of the β-diversity of the gut microbiota of black-spotted frogs showed an excellent model fit (stress = 0.089) (Figure 9). The confidence ellipses in the figure had an obvious overlap, from top to bottom, at the tillering, embryonic, full heading, and ripening stages of the rice. The results showed that the different densities of the rice–frog co-cropping models had no significant effect on the β-diversity of the gut microbiota of the black-spotted frogs. However, differences in the β-diversity of the gut microbiota of the frogs were noticed during the different growth stages of the rice.

#### 3.2.3. LEfSe Analysis of Differential Microbe Species

The LEfSe analysis of differential species in the guts of black-spotted frogs showed that the high-density rice–frog co-cropping group had more endemic microbe species (Figure 10), with 5 and 17 differentially significant species in the guts of the low-density and high-density rice–frog co-cropping groups, respectively (*p* < 0.05). The microbes that were significantly enriched in the high-density rice–frog co-crop group were 11_24_unclassified, *Sphingopyxis*, and Limnochordia_unclassified (*p* < 0.01).

#### 3.2.4. Predictive Analysis of Gut Microbial Function

STAMP differential analysis via PICRUSt2 functional prediction identified 22 differentially enriched KO genes in KEGG level 3 metabolic pathways at the rice ripening stage (Figure 11). On one hand, 12 KO-represented pathways were significantly enriched in the low-density co-cropping groups, including ‘two-component system’, ‘glutathione metabolism’, ‘atrazine degradation’, and neurological disease-related pathways (e.g., Huntington’s, Alzheimer’s, Parkinson’s disease), which had abundances higher than those of the high-density groups (*p* < 0.05). On the other hand, high-density co-cropping groups showed an elevated expression of the cysteine/methionine metabolism, ribosome, zeatin biosynthesis, and amino acid-related pathways (e.g., ‘aminoacyl-tRNA biosynthesis’) in the frog gut samples compared to the low-density groups (*p* < 0.05).

BugBase phenotypic analysis (Figure 12) showed a higher anaerobic bacterial abundance in the high-density co-cropping frog guts versus those of the low-density groups. Firstly, the low-density groups had elevated ‘contains mobile element’ genes at the tillering, embryonic, and ripening stages, but lower ‘facultative anaerobic’ genes compared to the high-density groups. Secondly, the ‘potentially pathogenic’ and ‘stress tolerant’ genes were more abundant in the low-density groups at the tillering, full heading, and ripening stages. Finally, Gram-positive bacteria dominated the low-density guts at the tillering and embryonic stages, while the high-density groups showed higher Gram-negative and biofilm-forming bacteria at the full heading and ripening stages.

### 3.3. Relationship Between Microbial Communities in Paddy Field Water and the Guts of Frogs

#### 3.3.1. Similarity Analysis

The species taxonomic comparisons between the groups were conducted after categorization based on the 100% similarity of the microbial ASVs was conducted. The paddy field water of the rice monoculture group, high-density rice–frog co-cropping group, and low-density rice–frog co-cropping group modes contained a large number of unique ASVs (7910, 9640, and 11,212 ASVs, respectively), which was much higher than the number of unique ASVs of the black-spotted frog’s guts from the low-density and high-density rice–frog co-cropping groups (1803 and 3117 ASVs, respectively) (Figure 13). Moreover, the number of unique ASVs in the guts of the black-spotted frogs in the high-density rice–frog co-cropping group (3117 ASVs) was also much higher than that in the low-density rice–frog co-cropping group (1803 ASVs). However, a total of 1797 ASVs were shared between the paddy field water in the three different cropping modes, and 574 ASVs were shared between the black-spotted frog guts from the two different frog culturing densities. In the high-density and low-density rice–frog co-cropping groups, 235 and 162 ASVs were shared between the gut microbes and the water microbes, respectively.

#### 3.3.2. Correlation Analysis of Water and Frog Gut Microbiota

To investigate the effect of the interaction between the water microbiota and frog gut microbiota, the top 15 genera from each at the ripening stage of the rice were selected for Spearman correlation analysis. The results showed that a total of 16 and 19 groups of genera were significantly correlated (*p* < 0.05) between the black-spotted frog gut microbiota and the paddy field water microbiota in the low-density rice–frog co-cropping group (Figure 14a) and high-density rice–frog co-cropping group (Figure 14b).

More specifically, in the low-density rice–frog co-cropping group, *Vibrio* and *Pseudoalteromonas*, as well as *Bradyrhizobium* and Gemmataceae_unclassified, showed highly significant positive correlations (*p* < 0.001). *Limnohabitans* and *Acinetobacter*, as well as *Ellin6067* and SC.1.84_unclassified, showed a significant positive correlation (*p* < 0.01). Betaproteobacteria_unclassified and Alphaproteobacteria_unclassified, as well as *Sphingomonas*, showed a positive correlation (*p* < 0.05), and *Limnohabitans* and *Sphingomonas* were positively correlated (*p* < 0.05). Acidobacteriales_unclassified as well as Alphaproteobacteria_unclassified and *Ellin6067* were positively correlated (*p* < 0.05). Alphaproteobacteria_unclassified and Others were positively correlated (*p* < 0.05). However, *Pseudoalteromonas* and *Vibrio* were negatively correlated with Betaproteobacteria_unclassified, Alphaproteobacteria_unclassified, and Others (*p* < 0.05).

In the high-density rice–frog co-cropping group, *Mycoplasma* was positively correlated with *Pseudoalteromonas* and *Vibrio* (*p* < 0.05), and there was a highly significant positive correlation between *Pseudoalteromonas* and *Vibrio* (*p* < 0.001). Additionally, a highly significant positive correlation was found between Gemmataceae_unclassified as well as *Bradyrhizobium* and Betaproteobacteria_unclassified (*p* < 0.01). *Sphingomona* was positively correlated with SC.1.84_unclassified, Acidobacteriales_unclassified, and Alphaproteobacteria_unclassified (*p* < 0.001, *p* < 0.01, *p* < 0.05, respectively). Chloroplast_unclassified, Alphaproteobacteria_unclassified, and Betaproteobacteria_unclassified were positively correlated (*p* < 0.001, *p* < 0.05, respectively). The results also showed that SC.I.84_unclassified was highly significantly positively correlated with Acidobacteriales_unclassified and *Ellin6067* (*p* < 0.001), while *Bacillus* and *Bradyrhizobium* showed a positive correlation (*p* < 0.05). Finally, *Ellin6067*, *Acinetobacter*, and *Limnohabitans* showed a positive correlation (*p* < 0.001, *p* < 0.01, respectively). Moreover, *Mycoplasma*, *Pseudoalteromonas*, *Vibrio*, and Others were also positively correlated (*p* < 0.001, *p* < 0.05, *p* < 0.05, respectively).

#### 3.3.3. Interrelationships Between Microbial Dominant Species

The microbial diversity of the paddy water was correlated with the dominant microbe species with a mean relative abundance >1% in the guts of the frogs (Figure 15).

There were positive correlations between all the dominant species of microbes in the paddy field water, except for *Sphingomonas*, Alphaproteobacteria_unclassified, SC-I-84, and Gemmataceae_unclassified, which were negatively correlated with most of the dominant species. Both the Shannon’s and Simpson’s indices of the gut microbiota of the black-spotted frogs were significantly correlated with *Limnohabitans* (r > 0.4, *p* < 0.05).

## 4. Discussion

### 4.1. Microbe Diversity in Rice Paddy Water from Different Cropping Patterns and Periods

Microbial communities are important components of water ecosystems, and microbiota mainly arrive from the air and soil, as well as plant and animal residues, secretions and excretions [38]. They play a vital role in the recycling and utilization of nitrogen, phosphorus, sulfur, and carbon in the water [39], and resultantly, the diversity and community structure of the water microbiota have a direct impact on the production capacity and ecological benefits of the system [40].

In this study, the microbiota were dominated by Proteobacteria in the paddy field water, with other high abundances of Bacteriota, Actinobacteria, and Firmicutes, which correlates with some previous studies [41,42,43]. The culturing density of black-spotted frogs did not significantly influence the α- and β-diversity indexes of the paddy field water microbiota; however, there was a significant difference in the microbial structure of the water during different rice growth periods. This may be because the microbial abundance in cultivated land or paddies depends on the cultivation practices and ecological parameters [44,45], and not inherent differences in the water, which had similar and relatively stable physicochemical environments and nutrients. Additionally, connectivity between the paddy field water and external water sources may lead to the spread of similar microbiota, and the similarity in the source of water among rice paddies may be one of the reasons why the microbial composition in different paddies tends to be similar [46]. Therefore, when farming practices are consistent between rice fields, it is difficult to influence the microbial diversity of the water on a larger scale, regardless of whether black-spotted frogs are cultured or not. However, during the growth of rice, the microbial diversity is dynamically regulated through multiple mechanisms, including root exudation patterns, carbon and nitrogen metabolism, and hormonal signaling [47,48,49].

There was a large number of differentially significant species in the water of the rice monoculture group and the rice–frog co-cropping groups. This may be because the rice monoculture group was in a single rice-growing ecosystem with a relatively homogeneous ecological niche, and root secretions mainly influence the formation and development of the microbial community despite paddy management practices (e.g., fertilizer application, irrigation, etc.). In contrast, the rice–frog co-cropping group was in a relatively complex ecosystem in which the presence of frogs introduced new biological factors [50]. Similarly, frog activities (e.g., predation, excretion, etc.) could change the physicochemical properties and biological composition of the water, leading to possible differences between the rice monoculture group and the rice–frog co-cropping group in environmental factors such as the dissolved oxygen, temperature, and pH in the water. A previous study showed that frog-aggregation soil exhibited markedly increased sulfur respiration and hydrocarbon degradation [26]. It has also been hypothesized that frog feces and skin carry a large number of microorganisms that enter the water column directly or indirectly, so frog behavior affects the microbial composition of the water [16,51].

In the rice monoculture group, enriched microbes such as Nitrosomonadaceae_unclassified interact with the rice root system, promoting rice growth. Other microbes such as *Bosea*, NS9_marine_group_unclassified, and Burkholderiales_unclassified may occupy specific ecological niches in this ecosystem, forming a relatively stable relationship with their surroundings. However, Burkholderiales, as a source of causative agents for melioidosis, may pose a potential threat to rice growth and human health, especially when the environmental conditions are suitable [52]. In addition, it has been found that some genera of Burkholderiales may be the original environmental hosts of the II trimethoprim resistance genes (*dfrB*) [53], and the presence of these microbes in rice monoculture groups may lead to the spread of resistance genes to other bacteria via horizontal gene transfer. This can increase bacterial resistance in the environment, with potential risks to future disease treatment and ecological health.

In the low-density rice–frog co-cropping group, the existence of *Xenorhabdus* may be related to the presence of frogs. As a symbiotic bacterium of entomopathogenic nematodes, *Xenorhabdus* may participate in controlling these pests in rice fields [54,55], and by taking advantage of this, the use of chemical pesticides can be reduced or abandoned, which would result in organic farming systems. Meanwhile, frog predation likely influences microbial communities, creating niches favorable for *Xenorhabdus*. On the other hand, rice monoculture systems may rely more on chemical pest control, posing environmental risks [14]. Thus, future studies should assess whether *Xenorhabdus* impacts frog health or the ecosystem balance.

The PICRUSt2 functional prediction difference analysis showed that the functional genes significantly enriched in the rice–frog co-cropping group and the rice-monocropping group were quite different. This suggests that the rice–frog co-cropping group promoted amino acid synthesis and metabolism in the paddy water, while also inhibiting some harmful bacteria in the paddy water. In addition, this study found that microbes related to elemental movement and stress resistance were significantly enriched in the water of the rice–frog co-cropping group during the embryonic, full heading, and ripening stages of the rice. This suggests that the rice–frog co-cropping model may be able to improve the element cycling and stress resistance of paddy ecosystems to a certain extent. These phenomena may be related to the activities and excretion of the black-spotted frog, and have contributed to improved water fertility in paddy fields and the better growth of rice. However, the potential pathogenicity of microbiota in the water of the high-density rice–frog co-cropping group was significantly higher at the embryonic, full heading, and ripening stages of the rice, possibly due to the high density of frogs. Additionally, studies in aquaculture have shown that a high rearing density can increase the potential pathogenicity [56], but no studies have found that the rice–frog co-cropping model improves the resilience of paddy aquatic ecosystems.

Therefore, rice–frog co-cropping affects the microbial diversity and community structure of paddy water in two main ways. Firstly, a large number of microbes in the intestinal tract of the black-spotted frog enter the water through its feces, enriching the microbial diversity of the water and affecting its community structure. Secondly, the activities of the black-spotted frog stir up the soil and promote the exchange of materials between the soil, the water environment, and the rice roots, which makes it easier for microbes in the soil to enter the water and thus affects the diversity and community structure of the water.

### 4.2. Gut Microbiota Diversity of Frogs and Differences Between Cropping Patterns and Periods

Gut microbes are closely linked to host pathogen defense [19]. Proteobacteria and Firmicutes were the most abundant microbes in the frog’s gut, which is consistent with the results of previous studies [57,58]. Some microbes in Firmicutes help the host to digest and absorb nutrients, while Proteobacteria are metabolically diverse and can adapt to a wide range of environmental conditions [45]. Proteobacteria enrichment in the frog’s gut during the full heading stage of rice may result from the dietary introduction of pollen or plant secretions, which potentially disrupts gut microbiota homeostasis [59,60]. As the rice grows, the black-spotted frogs adapt and develop their dietary habits, and, thus, the frog’s gut microbes may also vary according to the rice’s developmental stage [24]. In addition, the paddy field environment may change with the rice growth stage, water temperature, and water quality during different culture periods, which may affect the composition of the gut microbiota of the black-spotted frog [27].

*Vibrio* are widely distributed in nature, especially in water, and include *Vibrio cholerae* and *Vibrio parahaemolyticus*, which are common pathogenic microbiota in aquaculture [61]. In the current study, the guts of frogs in the low-density rice–frog co-cropping pattern had a higher abundance of *Vibrio*, and this suggests that appropriately increasing the frog culturing density might reduce the risk of bacterial pathogenicity. Moreover, *Cetobacterium* was significantly enriched in the guts of frogs in the high-density rice–frog co-cropping group. This is a beneficial symbiotic bacterium that is commonly found in the guts of fish and that can regulate the microbial composition of the guts, improve liver health, and inhibit viral and bacterial infections [62,63,64]. *Lactococcus* and *Cetobacterium* are the resident bacteria in the gut of the black-spotted frog. Moreover, *Pseudoalteromonas* and *Vibrio* may be related to the carriage of microbiota in feed and the influence of environmental microbiota. These two microbial groups were more abundant in the guts of frogs in the low-density group at the embryonic and full heading stages of the rice. In contrast, *Pseudoalteromonas* and *Vibrio* increased in the high-density group during the embryonic and ripening stages of the rice, which was possibly driven by interactions among the black-spotted frogs under high-density situations that altered the environmental microbiota and made it easier for these microbes to colonize and reproduce in the gut. Meanwhile, *Neochlamydia*, a predominantly free-living amoeba that can infect fish, was found in the guts during the later growth stages of rice [65]. Although it has been demonstrated that fish can be fully acclimatized to the presence of chlamydia [65], no studies have been conducted to determine whether it affects black-spotted frogs.

The α- and β-diversity indices of the gut microbiota of the frogs were unaffected by the frog density and showed significant different across the different periods of rice growth. A higher α-diversity index indicated greater microbial richness and evenness in the frog guts. Meanwhile, a higher α-diversity index also implies that the gut microbes are more resistant to interference, which promotes the stability of the intestinal environment and helps the host to better adapt to the external environment [66]. It has been demonstrated that the gut microbial structure of amphibians changes with maturity, which may be related to changes in their activity patterns, physiological needs, feeding behavior, and ecology [24]. Although this study tested three rice cropping modes with different densities of black-spotted frogs, these density differences may not have been sufficiently large to significantly alter the structure of the frogs’ gut microbes, as the frogs’ gut has a certain self-regulation and selection mechanism to maintain a relatively stable microbial structure.

In the high-density rice–frog co-cropping group, the frog guts harbored more endemic species, likely due to the model’s complex ecological network fostering tighter symbiotic relationships. Interestingly, some studies have shown that *Sphingopyxis* has bioremediation potential [67], and thus its presence in the high-density rice–frog co-cropping group may have a degradation and remediation effect on pesticide residues, heavy metals, and other pollutants in the paddy field and the surrounding environment, reducing the environmental pollution. In addition, these endemic microbes may spread to the environment around the paddy field as a result of the activities of the frogs, impacting and altering the microbial community structure in the surrounding soil and water and promoting microbial dispersal and exchange. Some genera in Limnochordia may have a significant impact on the microbial ecological characteristics of the surrounding ecosystems due to their participation in specific material cycling processes, such as biofilm formation and metabolic exchange [68]. Thus, the enrichment of Limnochordia in the frog intestine may help the black-spotted frog to enhance its intestinal microecosystem stability for improved health.

The results of the PICRUSt2 functional prediction difference analysis showed that the functional genes significantly enriched in the low-density rice–frog co-cropping group were mainly involved in amino acid metabolism and antibiotic synthesis. Similarly, the microbiota of the low-density rice–frog co-cropping group had a certain inhibitory effect on the harmful bacteria of the intestinal microbiota of the black-spotted frog and could help to degrade toxic substances in the environment, such as ethylbenzene and Atrazine. In contrast, a large number of pathogenic bacteria present in the low-density group were not detected in the high-density rice–frog co-crop group, with only microbiota related to amino acid synthesis being significantly enriched. In addition, the results of the BugBase phenotypic prediction also showed that the low-density rice–frog co-cropping group was significantly more resistant to stress than the high-density rice–frog co-cropping group, despite the presence of potentially pathogenic bacteria at the tillering, full heading, and ripening stages of the rice. This suggests that the reasonable placement of black-spotted frogs in paddy fields may promote the utilization of nutrients and the degradation of harmful substances in paddy fields to a certain extent, as well as improving the stress resistance of black-spotted frogs, even in the presence of some pathogenic bacteria.

### 4.3. Effects of Rice–Frog Co-Cropping Patterns on the Relationship Between Water Microbiota and Frog Gut Microbiota

The culture environment and the structure of gut microbiota are closely associated with the development of diseases in aquatic animals [38]. The composition and structure of the gut microbiota of aquatic animals can be influenced by the microbiota of the aquaculture environment, thus affecting their health. The gut microbiota and aquaculture microbiota are closely linked and constitute an organic whole, and environmental stresses that alter environmental microbes and affect microbial relationships can lead to dysbiosis of the host gut microbes, resulting in disease [21]. Interactions between different microbiota not only have an impact on the water environment, but also change the nature of microbial interactions [69]. Conversely, the water quality is one of the important factors that regulates microbial community activity [70].

In this study, the similarities and differences between the gut microbiota and environmental microbiota of black-spotted frogs at different culture densities were compared. The results indicated that the frog gut samples from the high-density rice–frog co-cropping group shared many ASVs with the water microbial samples and that the water microbiota contained a large number of endemic ASVs. This suggests that there is a close connection and interaction between the frog’s gut and the water microbiota. Frogs live in the water environment of paddy fields and inevitably come into contact with microbiota in the water. These shared ASVs may enter and colonize the frog’s gut through swallowing, body surface contact with the water, and other pathways. Furthermore, there are a variety of microenvironments in the water, such as different water layers, underwater sediments, and aquatic plant surfaces, and these microenvironments provide suitable living conditions for different types of microbiota, which leads to an overall high diversity of microbiota in the water [71].

Microbial diseases are among the most important diseases of aquaculture animals [72]. However, except for a few reports that describe the macrobiotic relationships between the aquatic environment and the aquatic animal gut, the interactions between the environmental microbiota and the gut microbiota of cultured amphibian species remain understudied [73]. In this study, the relationship between the frog gut and water microbiotas in a rice–frog co-cropping system was described for the first time, and the results showed that the water microbiota strongly influenced the gut microbiota of the black-spotted frog. Highly significant correlations were found between the gut microbiota of 19 and 22 bacterial genera and the water microbiota in the high-density and low-density frog groups, respectively, with a positive correlation being observed between most of the genera. What is more, regardless of the density of black-spotted frog placement, the *Vibrio* in the paddy water was highly significantly and positively correlated with *Pseudoalteromonas* in the intestinal tract, and less so with Betaproteobacteria_unclassified and *Bradyrhizobium*. This suggests that these microbiotas act synergistically and can assist in maintaining the stability of the paddy field water and the gut microbiota of black-spotted frogs. Several pathogenic *Vibrio* species and strains are recognized as important pathogens of exogenous infections [23,74]. For example, *Vibrio* cholerae is the pathogen that causes cholera, which can lead to severe diarrhea and dehydration in humans [75]. In addition, the presence of *Acinetobacter* in water may cause diseases such as intestinal inflammation, with symptoms including ulcers, bleeding, and increased mucus production [76]. On the other hand, *Pseudoalteromonas* can produce a variety of bioactive substances such as extracellular polysaccharides, antibiotics, and enzymes, which have potential applications in biocontrol, drug development, and industrial production [77,78]. In paddy water, *Pseudoalteromonas* can promote the fixation of nitrogen from the air, synthesize plant-available nitrogen compounds, provide nitrogen nutrition to plants, and promote rice growth [79].

Among the dominant species of water microbes, *Sphingomonas*, Alphaproteobacteria_unclassified, SC-I-84, and Gemmataceae were negatively correlated with most of the other dominant species. This may mean that they compete with other dominant species in the aquatic environment, whereas the other dominant species may have synergistic effects or common adaptations to similar environmental conditions in the aquatic environment. The genus *Limnohabitans*, a typical freshwater planktonic microorganism, is thought to play an important role in inland freshwater habitats [80,81]. The Shannon and Simpson indices are important indicators of microbial diversity and showed a significant correlation with *Limnohabitans* in the current study, suggesting that *Limnohabitans* may play an important role in the formation and maintenance of the diversity of the gut microbiota in black-spotted frogs, as well as in maintaining the microbial homeostasis of rice fields.

## 5. Conclusions

In this study, the gut microbiota of black-spotted frogs and the microbial structure of paddy water in rice–frog co-cropping models were analyzed with 16S high-throughput sequencing. It was the rice growth stage, not the frog density, that most strongly influenced the microbial diversity in both the gut and water, with higher diversity being seen in later growth stages. The paddy water harbored richer microbial communities than the frog guts, with shared dominant taxa (e.g., *Limnohabitans*) and specific niche variations. It is worth noting that low-density co-cropping promoted *Xenorhabdus* enrichment, and thus potentially enhanced entomopathogenic nematode activity for pest control, while showing more stable gut microbiota. Finally, a small number of pathogenic bacteria were found in the paddy field water and the guts of frogs in this study, although they have not been proven pathogenic to black-spotted frogs.

However, this study has limitations. For instance, relying solely on 16S rRNA sequencing limits the functional insights, and the single-season field trial restricts this study’s long-term validity. At the same time, changes in some environmental factors (e.g., temperature, pH) in the field experiment may affect the experiment to some extent. What is more, the pathogenic bacteria detected in the samples were not experimentally validated for their virulence. Thus, future studies should employ metagenomics to explore microbial functions, conduct multi-seasonal trials, and validate pathogen impacts through infection assays. This will enhance disease control and advance sustainable rice–frog co-cropping.

## Figures and Tables

**Figure 1 microorganisms-13-01700-f001:**
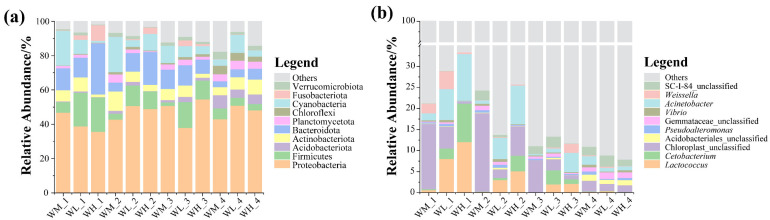
The relative abundance of the dominant microbes (top 10) in paddy field water at different levels: (**a**) relative abundance of bacteria at the phylum level; (**b**) relative abundance of bacteria at the genus level. WM, paddy field water of rice monoculture group; WL, paddy field water of low-density rice–frog co-cropping group; WH, paddy field water of high-density rice–frog co-cropping group; 1, 2, 3, and 4 represent the tillering, embryonic, full heading, and ripening stages of rice, respectively.

**Figure 2 microorganisms-13-01700-f002:**
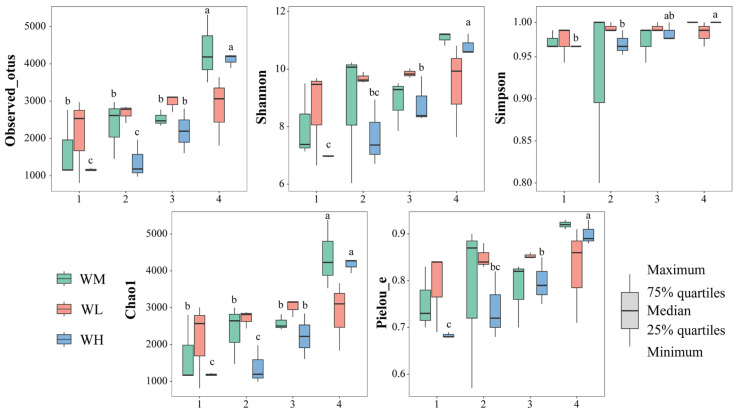
The microbial α-diversity indices of the paddy field water (n = 3). WM, paddy field water of rice monoculture group; WL, paddy field water of low-density rice–frog co-cropping group; WH, paddy field water of high-density rice–frog co-cropping group; 1, 2, 3, and 4 represent the tillering, embryonic, full heading, and ripening stages of rice, respectively. Different letters indicate significant differences obtained via Tukey’s test (passed Levene’s test for homogeneity) or Dunnett’s T3 test (do not pass Levene’s test for homogeneity), significance level alpha = 0.05.

**Figure 3 microorganisms-13-01700-f003:**
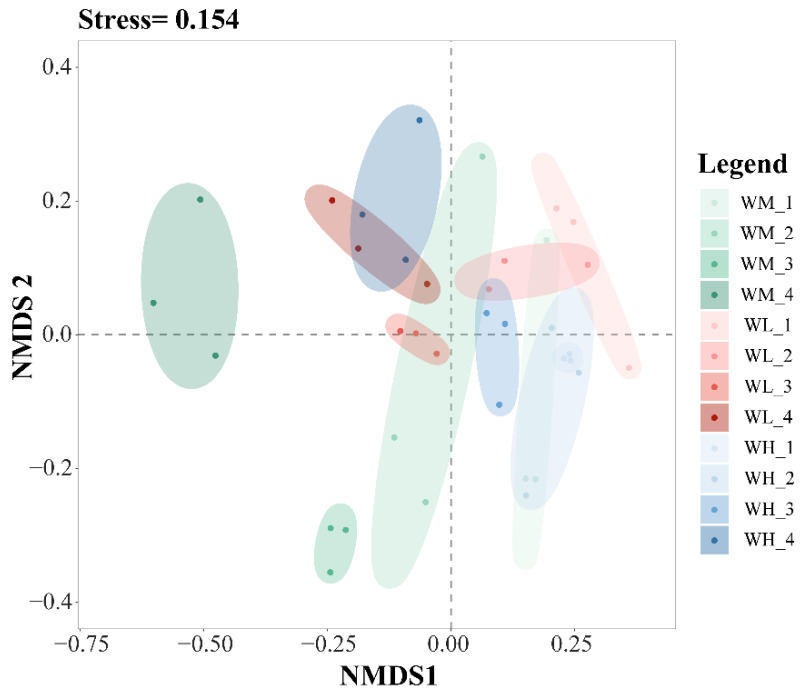
An NMDS analysis of the paddy field water microbiota based on the Bray–Curtis distance (n = 3). WM, paddy field water of rice monoculture group; WL, paddy field water of low-density rice–frog co-cropping group; WH, paddy field water of high-density rice–frog co-cropping group; 1, 2, 3, and 4 represent the tillering, embryonic, full heading, and ripening stages of rice, respectively. The outer circles indicate 95% confidence intervals.

**Figure 4 microorganisms-13-01700-f004:**
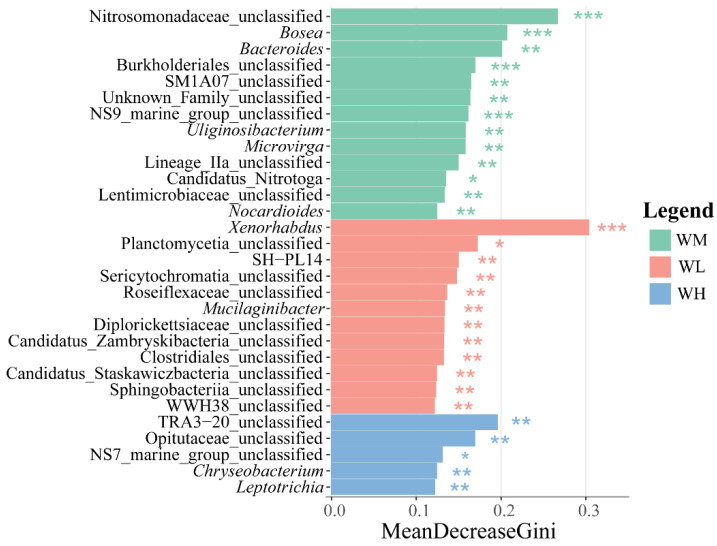
Distribution and significance of microbial LDA values in the paddy field water (n = 3). LEfSe-identified differentially abundant taxa with LDA > 2.0 and FDR < 0.05. Bar width represents the LDA score. WM, paddy field water of rice monoculture group; WL, paddy field water of low-density rice–frog co-cropping group; WH, paddy field water of high-density rice–frog co-cropping group. *, *p* < 0.05; **, *p* < 0.01; ***, *p* < 0.001, “*p*” is the Wilcoxon test with correction by BH procedure.

**Figure 5 microorganisms-13-01700-f005:**
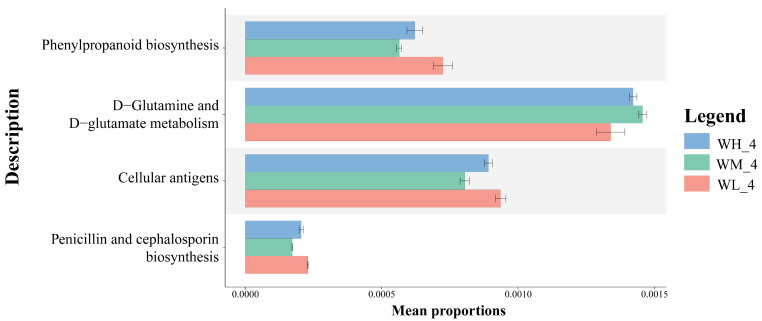
Annotation of functional genes (mean ± S.D.) in KEGG layer 3 of microbiota isolated from rice paddy fields (n = 3). WM, paddy field water of rice monoculture group; WL, paddy field water of low-density rice–frog co-cropping group; WH, paddy field water of high-density rice–frog co-cropping group; 4 represents the ripening stage of rice. Pictures are displayed for the functions with a *p*-value < 0.05 in the *t*-test difference test results for two-by-two comparisons.

**Figure 6 microorganisms-13-01700-f006:**
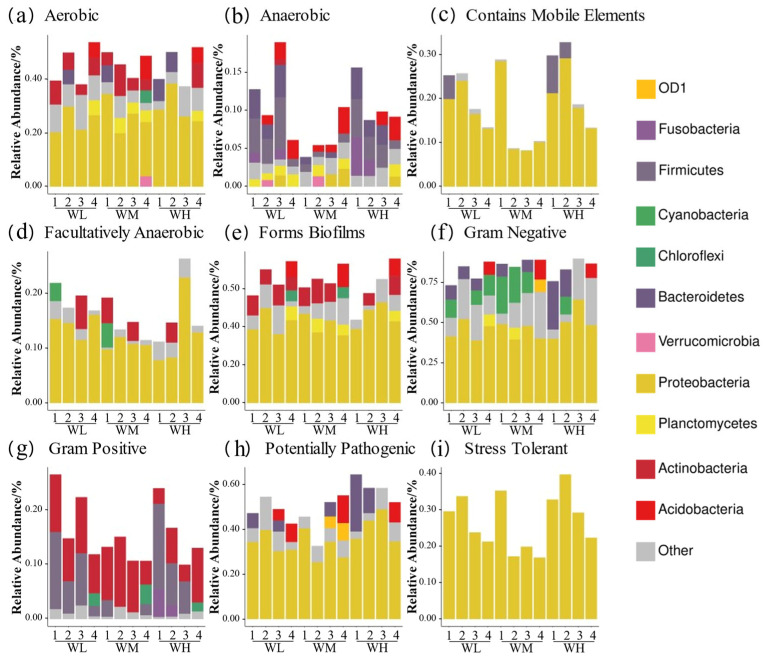
Predicted BugBase phenotypes of rice paddy field water microbiota in China: (**a**) aerobic; (**b**) anaerobic; (**c**) contains mobile elements; (**d**) facultatively anaerobic; (**e**) forms biofilms; (**f**) Gram-negative; (**g**) Gram-positive; (**h**) potentially pathogenic; (**i**) stress tolerant. WM, paddy field water of rice monoculture group; WL, paddy field water of low-density rice–frog co-cropping group; WH, paddy field water of high-density rice–frog co-cropping group; 1, 2, 3, and 4 represent the tillering, embryonic, full heading, and ripening stages of rice, respectively.

**Figure 7 microorganisms-13-01700-f007:**
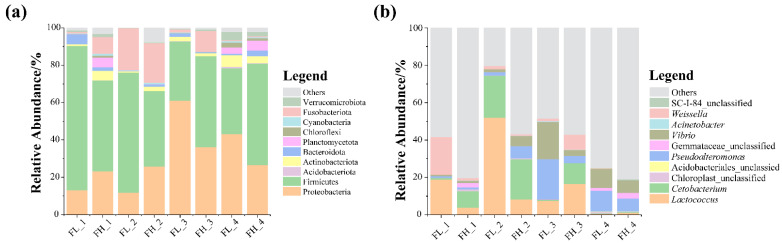
The relative abundance of dominant microbes (top 10) in the guts of black-spotted frogs at different levels: (**a**) relative abundance of bacteria at the phylum level; (**b**) relative abundance of bacteria at the genus level. FL, gut samples of frogs in low-density rice–frog co-cropping group; FH, gut samples of frogs in high-density rice–frog co-cropping group; 1, 2, 3, and 4 represent the tillering, embryonic, full heading, and ripening stages of rice, respectively.

**Figure 8 microorganisms-13-01700-f008:**
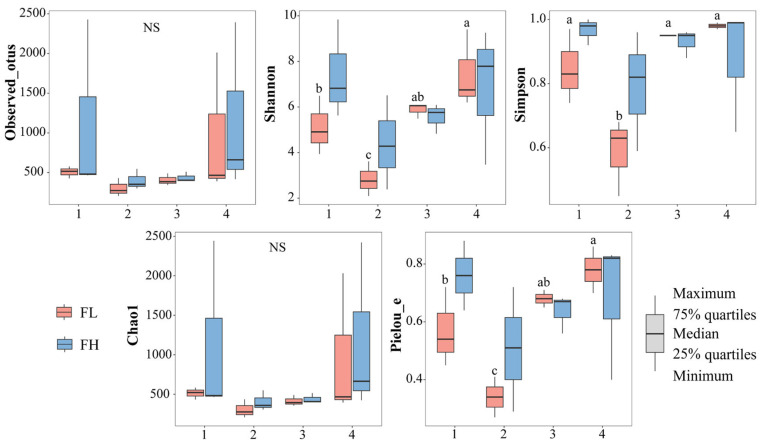
Differences in the α-diversity indices of the gut microbiota of black-spotted frogs in rice paddies in China (n = 3). FL, gut samples of frogs in low-density rice–frog co-cropping group; FH, gut samples of frogs in high-density rice–frog co-cropping group; 1, 2, 3, and 4 represent the tillering, embryonic, full heading, and ripening stages of rice, respectively. Different letters indicate significant differences via Tukey’s test (passed Levene’s test for homogeneity) or Dunnett’s T3 test (did not pass Levene’s test for homogeneity), significance level alpha = 0.05.

**Figure 9 microorganisms-13-01700-f009:**
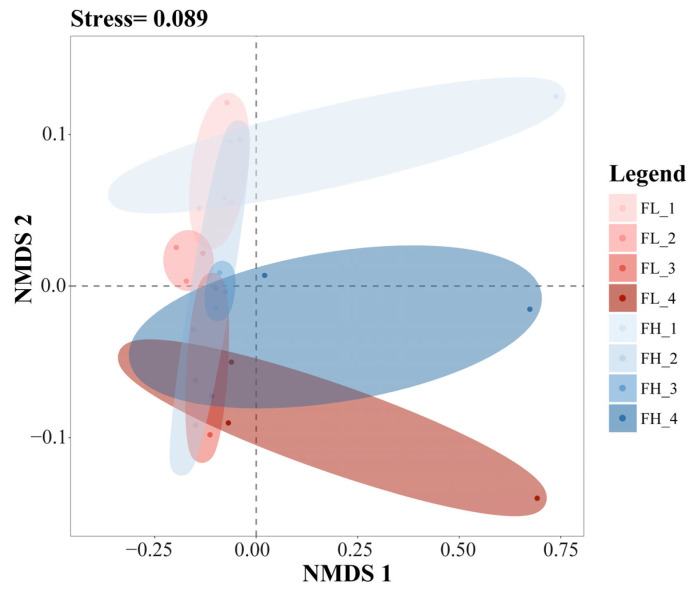
The NMDS analysis of the gut microbiota of black-spotted frogs from rice paddies in China based on the Bray–Curtis distance (n = 3). FL, gut samples of frogs in low-density rice–frog co-cropping group; FH, gut samples of frogs in high-density rice–frog co-cropping group; 1, 2, 3, and 4 represent the tillering, embryonic, full heading, and ripening stages of rice, respectively. The outer circles indicate 95% confidence intervals.

**Figure 10 microorganisms-13-01700-f010:**
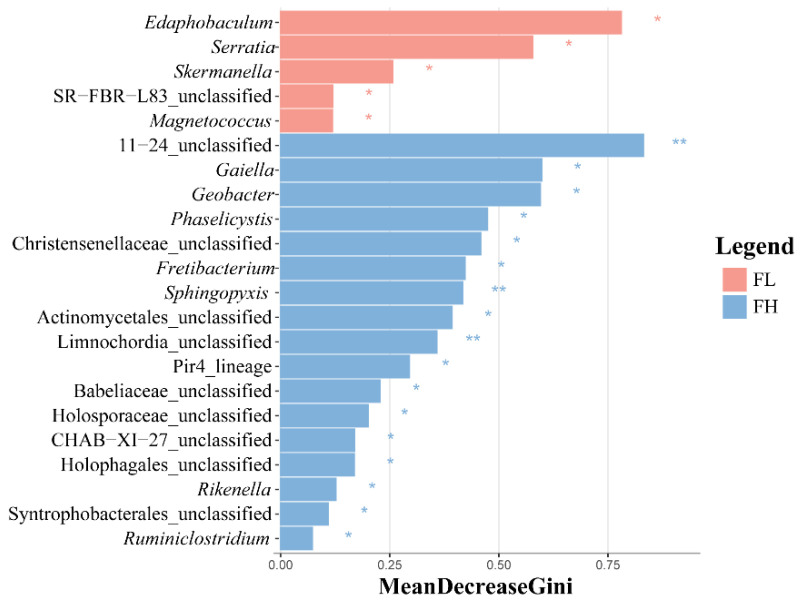
Distribution and significance of LDA values of gut microbes of the black-spotted frog from rice paddies in China (n = 3). LEfSe-identified differentially abundant taxa with LDA > 2.0 and FDR < 0.05. Bar width represents the LDA score. FL, gut samples of frogs in low-density rice–frog co-cropping group; FH, gut samples of frogs in high-density rice–frog co-cropping group; 1, 2, 3, and 4 represent the tillering, embryonic, full heading, and ripening stages of rice, respectively. *, *p* < 0.05; **, *p* < 0.01. “*p*” is the Wilcoxon test with correction by BH.

**Figure 11 microorganisms-13-01700-f011:**
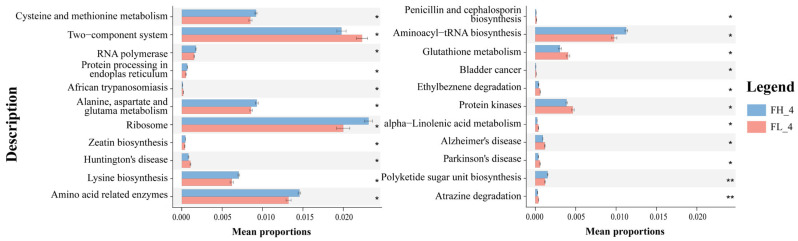
Annotation of gut microbial functional genes (mean ± S.D.) in KEGG level 3 of black-spotted frogs in rice paddies in China (n = 3). FL, gut samples of frogs in low-density rice–frog co-cropping group; FH, gut samples of frogs in high-density rice–frog co-cropping group; 4 represents the ripening stage of rice. Pictures are displayed for the functions with a *p*-value < 0.05 in the *t*-test difference test results for two-by-two comparisons. *, *p* < 0.05; **, *p* < 0.01.

**Figure 12 microorganisms-13-01700-f012:**
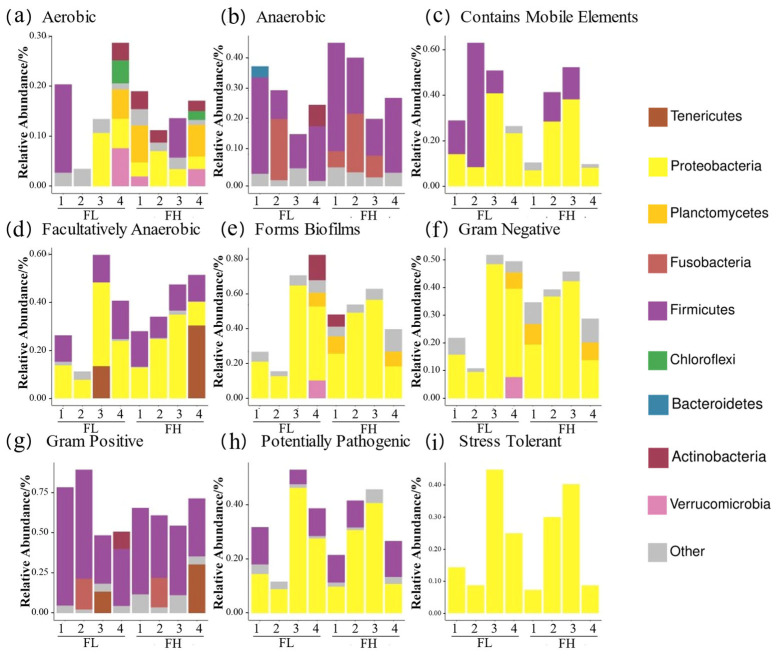
Predicted BugBase phenotypes of the gut microbes of black-spotted frogs in rice paddies in China: (**a**) aerobic; (**b**) anaerobic; (**c**) contains mobile elements; (**d**) facultatively anaerobic; (**e**) forms biofilms; (**f**) Gram-negative; (**g**) Gram-positive; (**h**) potentially pathogenic; (**i**) stress tolerant. FL, gut samples of frogs in low-density rice–frog co-cropping group; FH, gut samples of frogs in high-density rice–frog co-cropping group; 1, 2, 3, and 4 represent the tillering, embryonic, full heading, and ripening stages of rice, respectively.

**Figure 13 microorganisms-13-01700-f013:**
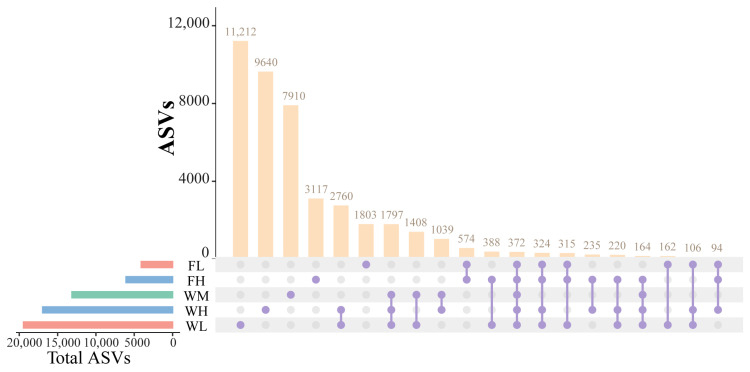
Similarity analysis of water and frog gut microbiota from rice paddies in China. WM, paddy field water of rice monoculture group; WL, paddy field water of low-density rice–frog co-cropping group; WH, paddy water of high-density rice–frog co-cropping group; FL, gut sample of frogs in low-density rice–frog co-cropping group; FH, gut sample of frogs in high-density rice–frog co-cropping group.

**Figure 14 microorganisms-13-01700-f014:**
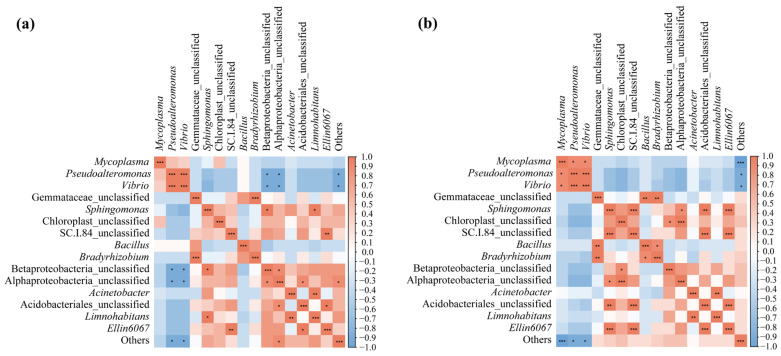
Correlation analysis of the paddy field water microbiota and frog gut microbiota from two different frog culturing densities: (**a**) microbiotas from the low-density rice–frog co-cropping model; (**b**) microbiotas from the high-density rice–frog co-cropping model. *, *p* < 0.05; **, *p* < 0.01; ***, *p* < 0.001 for Pearson correlation.

**Figure 15 microorganisms-13-01700-f015:**
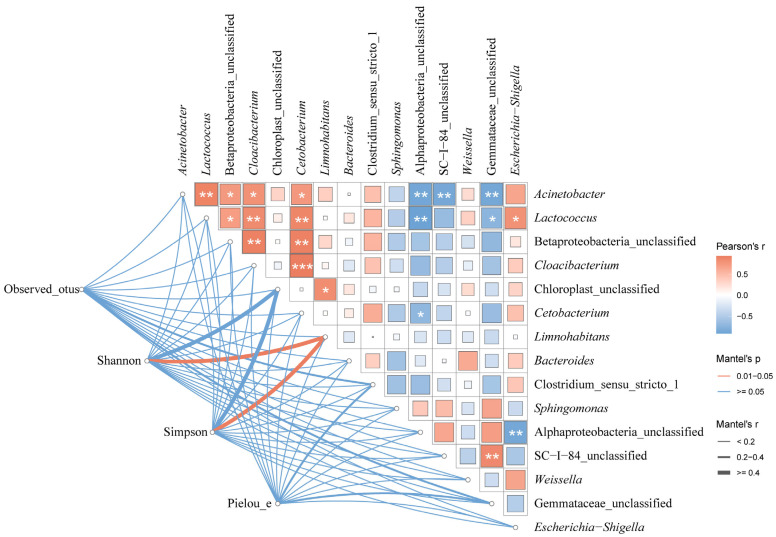
The relationship between the dominant microbe species in the paddy field water and the guts of black-spotted frogs in rice paddies in China. Edge widths indicate Mantel’s r statistics corresponding to distance correlations, and edge colors indicate statistical significance based on 9999 permutations, which show pairwise comparisons of environmental dominant species and gut microbial diversity. The color gradient in the box indicates Pearson’s correlation coefficient, the magnitude of which is proportional to the r-value, and the number indicates Pearson’s r-value. *, *p* < 0.05; **, *p* < 0.01; ***, *p* < 0.001 for Pearson correlation.

**Table 1 microorganisms-13-01700-t001:** Primers used for the 16S amplification of microbial samples from frogs and rice paddies in China.

Primer Name	Nucleotide Sequence (5′-3′)	Application
338F	ACTCCTACGGGGAGGCAGCA	Amplification of the V3-V4 regions of the 16S rRNA gene
806R	GGACTACHVGGGTWTCTAAT	

**Table 2 microorganisms-13-01700-t002:** Parameters of one-way ANOVA for the microbial α-diversity indices of the paddy field water (n = 3).

Items	Observed_otus			Shannon			Simpson			Chao1			Pielou_e		
Group	WM	WL	WH	WM	WL	WH	WM	WL	WH	WM	WL	WH	WM	WL	WH
df	3 (between-groups), 8 (within-groups) for all														
F	6.457	0.780	32.398	2.580	0.660	15.924	0.664	0.893	6.152	6.379	0.774	31.941	1.564	0.536	12.105
*p*	0.016	0.543	<0.001	0.126	0.599	<0.001	0.597	0.485	0.018	0.162	0.540	<0.001	0.272	0.671	0.002

WM, paddy field water of rice monoculture group; WL, paddy field water of low-density rice–frog co-cropping group; WH, paddy field water of high-density rice–frog co-cropping group. F, F-value; *p*, *p*-value.

**Table 3 microorganisms-13-01700-t003:** Parameters of one-way ANOVA for the microbial α-diversity indices of black-spotted frogs in rice paddies in China (n = 3).

Items	Observed_otus		Shannon		Simpson		Chao1		Pielou_e	
Group	FL	FH	FL	FH	FL	FH	FL	FH	FL	FH
df	3 (between-groups), 8 (within-groups) for all									
F-value	1.151	0.854	8.454	1.195	13.622	0.916	1.144	0.850	13.772	1.082
*p*-value	0.386	0.503	0.007	0.372	0.002	0.475	0.389	0.505	0.002	0.410

WM, paddy field water of rice monoculture group; WL, paddy field water of low-density rice–frog co-cropping group; WH, paddy field water of high-density rice–frog co-cropping group.

## Data Availability

The original contributions presented in this study are included in the article. Further inquiries can be directed to the corresponding author.

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
