# Peer review of "Frog Density and Growth Stage of Rice Impact Paddy Field and Gut Microbial Communities in Rice–Frog Co-Cropping Models"

_microorganisms, 2025, doi:10.3390/microorganisms13071700_

Round 1
Reviewer 1 Report
Comments and Suggestions for Authors
The study presents valuable insights into the interactions between frog gut microbiota and paddy field water microbiota under different rice-frog co-cropping models. The experimental design is generally appropriate, and the integration of high-throughput sequencing with microbial diversity and functional analyses is commendable. However, there are several issues regarding clarity, methodological rigor, and data interpretation that require attention.

Author Response
The study presents valuable insights into the interactions between frog gut microbiota and paddy field water microbiota under different rice-frog co-cropping models. The experimental design is generally appropriate, and the integration of high-throughput sequencing with microbial diversity and functional analyses is commendable. However, there are several issues regarding clarity, methodological rigor, and data interpretation that require attention.
Response: We appreciate the reviewer for acknowledging the study's value and providing constructive comments. We have carefully addressed the concerns regarding clarity, methodological rigor, and data interpretation. Firstly, we have refined some of the areas that were not clearly articulated for the reader's convenience. Secondly, we realized the omissions in the Materials and Methods section and added to them. In addition, we have added some recent literature and replaced some of it. Finally, in the Discussion section, we have simplified the statements and removed some sentences that may cause misunderstanding.
- The rationale for selecting the frog stocking densities (5000 and 10,000 frogs/mu) is not sufficiently justified. Please clarify how these densities relate to standard or ecologically relevant practices, and expand on why frog density might influence microbial diversity in the introduction section.
Response: We thank you for pointing out imperfections and providing suggestions, which have been added to the manuscript as described below.
- The reasons for selecting the frog stocking densities
The selection of stocking density of black-spotted frogs is based on some relevant basic research(Fang et al., 2021; Zheng et al., 2019). On the one hand, experiment has shown that black-spotted frogs are placed at densities of 7~15 frogs/m2 with higher survival rates and benefits (Zheng et al., 2019). On the other hand, a previous study on rice-frog co-cropping has confirmed that the density of frogs required to achieve optimal ecological and economic benefits is 60,000 frogs/ha (6 frogs/m2) (Fang et al., 2021). In addition, we have also added a note at the appropriate place in the manuscript.
Zeng Tao, Xiong Yutang, Chen Zhuo, et al. Suitable frog rearing density for rice-frog symbiosis in rice areas of Guizhou. Guizhou Agricultural Science, 2019, 47(11): 84-87.
Fang K, Dai W, Chen H, Wang J, Gao H, Sha Z, Cao L. The effect of integrated rice-frog ecosystem on rice morphological traits and methane emission from paddy fields. Sci Total Environ. 2021, 20(783): 147123. doi: 10.1016/j.scitotenv.2021.147123.
In this study, “Yong You No. 31” rice and black-spotted frogs were selected as the rice and frog species, respectively, for the rice-frog co-cropping model. The frogs selected were healthy, disease-free, and weighed 1~5 g. To ensure treatment homogeneity, frogs were visually sorted by size (snout-vent length: 1.6~1.9 cm) and mass (1.5~2.0 g) at the start of the experiment. A pre-experiment t-test confirmed no significant mass difference between low (1.733 ± 0.218 g) and high (1.654 ± 0.265 g) density groups (n = 30, t = 1.254, p = 0.215). All frogs were disinfected in 2%~3% salt water for 5~10 minutes before stocking, and the same treatment was applied to all fields, with no pesticide application during the entire period. A previous study on rice-frog co-cropping has confirmed that the density of frogs required to achieve optimal ecological and economic benefits is 60,000 frogs/ha (6 frogs/m2) [30]. Since the weight of the tiger frog (Rana rugulosa Wiegmann, 1834) used in this study (15 g on average) was much higher than that of the black-spotted frog (P. nigromaculatus Hallowell, 1861) used in the present experiment (1-5 g), an increase in frog density was appropriate for this experiment, the density of frogs was increased in this experiment. (Refer to the first paragraph of the section “2.2. Sample collection and processing”, L118~132)
Added reference:
30 Fang K, Dai W, Chen H, Wang J, Gao H, Sha Z, Cao L. The effect of integrated rice-frog ecosystem on rice morphological traits and methane emission from paddy fields. Sci Total Environ. 2021, 20(783): 147123. doi: 10.1016/j.scitotenv.2021.147123.
- Reasons why frog density may affect microbial diversity
We have already explained in the introduction section how frog density may affect microbial diversity.
Similarly, paddy water is one of the most important surrounding environments for black-spotted frogs in the rice-frog co-cropping models [28,29], but the relationship between paddy water microbiota and the black-spotted frog gut microbiota has not been studied. In addition, changes in frog density may have an impact on microbial diversity by affecting the frequency of contact between individuals, competitive pressures, and so on, which in turn affects their health and ultimately their microbial diversity. However, there are few reports on the effects of density on the microbial structure of the frog's gut. (Refer to the fourth paragraph of the section “1. Introduction”, L98~104)
Added reference:
28 Lukanov, S.; Kolev, A.; Dimitrova, B.; Popgeorgiev, G. Rice Fields as Important Habitats for Three Anuran Species—Significance and Implications for Conservation. Animals 2023, 14, 106, doi:10.3390/ani14010106.
29 Li, B.; Zhang, W.; Wang, Z.; Xie, H.; Yuan, X.; Pei, E.; Wang, T. Effects of Landscape Heterogeneity and Breeding Habitat Diversity on Rice Frog Abundance and Body Condition in Agricultural Landscapes of Yangtze River Delta, China. Curr Zool 2020, 66, 615–623, doi:10.1093/cz/zoaa025.
- Several statements, particularly those relating to microbial functions or ecological roles, would benefit from additional citations to support the claims.
Response: Thank you for pointing out our shortcomings in citing the literature. We fully agree that the discussion of microbial functions and ecological roles needs to be supported by more adequate literature to enhance the reliability of the conclusions. To address this issue, we have systematically checked the full text and supplemented it with authoritative studies in relevant fields.
Added reference:
21 Khaledi, M.; Poureslamfar, B.; Alsaab, H.O.; Tafaghodi, S.; Hjazi, A.; Singh, R.; Alawadi, A.H.; Alsaalamy, A.; Qasim, Q.A.; Sameni, F. The Role of Gut Microbiota in Human Metabolism and Inflammatory Diseases: A Focus on Elderly Individuals. Ann Microbiol 2024, 74, 1, doi:10.1186/s13213-023-01744-5.
22 Dodd, D.; Spitzer, M.H.; Van Treuren, W.; Merrill, B.D.; Hryckowian, A.J.; Higginbottom, S.K.; Le, A.; Cowan, T.M.; Nolan, G.P.; Fischbach, M.A.; et al. A Gut Bacterial Pathway Metabolizes Aromatic Amino Acids into Nine Circulating Metabolites. Nature 2017, 551, 648–652, doi:10.1038/nature24661.
23 de Souza Valente, C.; Wan, A.H.L. Vibrio and Major Commercially Important Vibriosis Diseases in Decapod Crustaceans. J Invertebr Pathol 2021, 181, 107527, doi:10.1016/j.jip.2020.107527.
47 Zhou, H.Z.; Wang, B.Q.; Ma, Y.H.; Sun, Y.Y.; Zhou, H.L.; Song, Z.; Zhao, Y.; Chen, W.; Min, J.; Li, J.W.; et al. The Combination of Metagenomics and Metabolomics Reveals the Effect of Nitrogen Fertilizer Application Driving the Remobilization of Immobilization Remediation Cadmium and Rhizosphere Microbial Succession in Rice. J Hazard Mater 2025, 487, 137117, doi:10.1016/j.jhazmat.2025.137117.
48 Dong, H.; Sun, H.; Chen, C.; Zhang, M.; Ma, D. Compositional Shifts and Assembly in Rhizosphere-Associated Fungal Microbiota Throughout the Life Cycle of Japonica Rice Under Increased Nitrogen Fertilization. Rice 2023, 16, 34, doi:10.1186/s12284-023-00651-2.
49 Kim, B.; Westerhuis, J.A.; Smilde, A.K.; Floková, K.; Suleiman, A.K.A.; Kuramae, E.E.; Bouwmeester, H.J.; Zancarini, A. Effect of Strigolactones on Recruitment of the Rice Root-Associated Microbiome. FEMS Microbiol Ecol 2022, 98, fiac010 doi:10.1093/femsec/fiac010.
51 Sammon NB, Harrower KM, Fabbro LD, Reed RH. Microfungi in drinking water: the role of the frog Litoria caerulea. Int J Environ Res Public Health. 2010 ,7,3225-3234, doi: 10.3390/ijerph7083225. Epub 2010 Aug 19.
56 Ellison, A.R.; Uren Webster, T.M.; Rodriguez-Barreto, D.; de Leaniz, C.G.; Consuegra, S.; Orozco-terWengel, P.; Cable, J. Comparative Transcriptomics Reveal Conserved Impacts of Rearing Density on Immune Response of Two Important Aquaculture Species. Fish Shellfish Immunol 2020, 104, 192–201, doi:10.1016/j.fsi.2020.05.043.
- Please clarify the statistical procedures and thresholds used, including corrections for multiple comparisons.
Response: Thank you very much for your suggestion. We have added some of the necessary statistical procedures and thresholds in the data analysis section.
The ASV data were analyzed using QIIME 2 (version 2021.8) to calculate the alpha-diversity (α-diversity, including the Observed_otus, Shannon, Simpson, Chao1, and Pielou_e) metrics and to perform non-metric multidimensional scaling analysis (NMDS) of the beta-diversity (β-diversity) in the samples to explore changes in the microbial composition of the paddy water and the frog’s guts. With the use of SPSS 21.0 (IBM, Inc., Armonk, NY, USA), differences in the α-diversity indices of the samples from different experimental groups were tested using a one-way analysis of variance (ANOVA). Significant differences between periods and frog densities were also compared. The applicability of the data to one-way ANOVA was tested by using the Shapiro-Wilk test (p > 0.05). Multiple comparisons were performed using Tukey’s test or Dunnett’s T3 test depending on whether they passed Levene’s test for homogeneity.
The Silva database (Release 138) was used as a reference to annotate the feature sequences with precise taxonomy using a simple Bayesian classifier[34]. With the use of R 4.4.2, LEfSe was used to analyze the differences in the relative abundance of the microbiotas of the paddy water and the gut of frogs under different culture densities[35]. Taxa with significant p-values (p < 0.05) and LDA scores ≥ 3 were considered to be differentially abundant taxa [36]. Using the Wilcoxon test and correcting for FDR by the Benjamini-Hochberg (BH) procedure. The functional abundance of the samples was predicted in PICRUSt2 based on marker gene sequence abundance. (Refer to the first and second paragraphs of the section “2.5. Data analysis”, L186~204)
Added reference:
36 Li, Z.; Ni, M.; Yu, H.; Wang, L.; Zhou, X.; Chen, T.; Liu, G.; Gao, Y. Gut Microbiota and Liver Fibrosis: One Potential Biomarker for Predicting Liver Fibrosis. Biomed Res Int 2020, 2020, 3905130, doi:10.1155/2020/3905130.
Another part of the information we gave in the figure notes of the original manuscript (e.g., L357) may have caused some inconvenience in reading, so we also added the relevant content in the Materials and Methods section.
Based on the top 15 genera from the water microbiota and frog gut microbiota, Pearson’s correlation analysis was performed to assess the statistical significance of the differences between the water microbiota and frog gut microbiota. The relationships between the microbial diversity of paddy field water and dominant microbe species with a mean relative abundance > 1% in the guts of frogs in both low- and high-density rice co-cropping groups were analyzed by using Cramer’s V. Mantel test analysis (with 9999 permutations) [37]. Correlation coefficient r- and p-values were obtained from the Pearson’s correlation analysis of the dominant species in the paddy water and frog samples by the Mantel function. After that, the Mantel test was completed in the vegan and ggcor packages and then visualized and plotted. (Refer to the fourth paragraph of the section “2.5. Data analysis”, L210~219)
Added reference:
37 Yang, L.; Pan, R.; Wang, S.; Zhu, Z.; Li, H.; Yang, R.; Sun, X.; Ge, B. Macrofaunal Biodiversity and Trophic Structure Varied in Response to Changing Environmental Properties along the Spartina Alterniflora Invasion Stages. Mar Pollut Bull 2025, 214, 117756, doi:10.1016/j.marpolbul.2025.117756.
- The manuscript lacks detailed mechanistic explanations for observed patterns. For example: Why does Xenorhabdus enrichment occur in low-density systems? Is it linked to frog behavior or nutrient cycling? How do rice growth stages directly modulate microbial diversity? How frog activities specifically alter water chemistry/microbiota.
Response: Thank you for raising these critical questions regarding the mechanistic interpretations of our findings. We acknowledge that the manuscript currently lacks detailed explanations for the observed patterns, and we have addressed this by incorporating additional discussions and hypothetical frameworks in the revised version. Below is a structured response to each of your points, with specific plans for revision.
- Why does Xenorhabdus enrichment occur in low-density systems?
A previous study has revealed changes in the nutrient and trace element composition of rice in rice-frog co-cropping systems (Sha et al., 2017). It is possible that the response of microbial communities in rice-frog ecosystems to environmental changes may be responsible for the enrichment of Xenorhabdus in low-density rice-frog co-cropping systems. However, there is no direct study on the mechanism of Xenorhabdus enrichment in the low-density system, and we will further investigate in the future.
Sha Z, Chu Q, Zhao Z, Yue Y, Lu L, Yuan J, Cao L. Variations in nutrient and trace element composition of rice in an organic rice-frog coculture system. Scientific Reports 2017, 16; 7(1):15706. doi: 10.1038/s41598-017-15658-1. (This reference was cited in the original article, Refer to the 11th of the section “Reference”, L829~830)
- Is it linked to frog behavior or nutrient cycling?
The literature on Xenorhabdus is mostly concerned with the symbiotic relationship between Xenorhabdus and entomopathogenic nematodes and its ecological functions, but there are fewer articles on nutrient cycling and microbial enrichment. However, it has been demonstrated that the introduction of frogs into rice paddies can control pest populations and increase microbial biomass in the soil (Sha et al., 2017). Although Xenorhabdus was not directly studied, the effects of the rice-frog co-cropping system on soil nutrients and rice micronutrient content were described, providing a basis for analyzing the relationship between nutrient cycling and microbial enrichment. In the future, the ecological mechanism of Xenorhabdus enrichment in the low-density rice-frog co-cropping system can be further explored.
Sha Z, Chu Q, Zhao Z, Yue Y, Lu L, Yuan J, Cao L. Variations in nutrient and trace element composition of rice in an organic rice-frog coculture system. Scientific Reports 2017, 16; 7(1):15706. doi: 10.1038/s41598-017-15658-1.
- How do rice growth stages directly modulate microbial diversity?
I am very sorry that the original manuscript focused on explaining why density did not have an effect on microbial diversity and ignored how microbial diversity is directly regulated during the rice growth stage phase. Numerous studies have shown that microbial diversity is dynamically regulated during the growth phase of rice through multiple mechanisms including root exudation patterns, carbon and nitrogen metabolism, and hormonal signaling (Dong et al., 2023; Zhou et al., 2025; Kim et al., 2022). We have added the explanation to the manuscript.
Dong H, Sun H, Chen C, Zhang M, Ma D. Compositional Shifts and Assembly in Rhizosphere-Associated Fungal Microbiota Throughout the Life Cycle of Japonica Rice Under Increased Nitrogen Fertilization. Rice (New York, N. Y.). 2023, 1; 16(1): 34. doi: 10.1186/s12284-023-00651-2.
Zhou HZ, Wang BQ, Ma YH, Sun YY, Zhou HL, Song Z, Zhao Y, Chen W, Min J, Li JW, He T. The combination of metagenomics and metabolomics reveals the effect of nitrogen fertilizer application driving the remobilization of immobilization remediation cadmium and rhizosphere microbial succession in rice. Journal of Hazardous Materials. 2025; 487: 137117. doi: 10.1016/j.jhazmat.2025.137117.
Kim B, Westerhuis JA, Smilde AK, Floková K, Suleiman AKA, Kuramae EE, Bouwmeester HJ, Zancarini A. Effect of strigolactones on recruitment of the rice root-associated microbiome FEMS Microbiology Ecology. 2022, 8;98(2): fiac010. doi: 10.1093/femsec/fiac010.
Therefore, when farming practices are consistent between rice fields, it is difficult to influence the microbial diversity of the water on a larger scale, regardless of whether black-spotted frogs are cultured or not. But during the growth of rice, microbial diversity is dynamically regulated through multiple mechanisms including root exudation patterns, carbon and nitrogen metabolism, and hormonal signaling [47–49]. (Refer to the second paragraph of the section “4.1. Microbe diversity in rice paddy water from different cropping patterns and periods”, L557~562)
Added reference:
47 Zhou, H.Z.; Wang, B.Q.; Ma, Y.H.; Sun, Y.Y.; Zhou, H.L.; Song, Z.; Zhao, Y.; Chen, W.; Min, J.; Li, J.W.; et al. The Combination of Metagenomics and Metabolomics Reveals the Effect of Nitrogen Fertilizer Application Driving the Remobilization of Immobilization Remediation Cadmium and Rhizosphere Microbial Succession in Rice. J Hazard Mater 2025, 487, 137117, doi:10.1016/j.jhazmat.2025.137117.
48 Dong, H.; Sun, H.; Chen, C.; Zhang, M.; Ma, D. Compositional Shifts and Assembly in Rhizosphere-Associated Fungal Microbiota Throughout the Life Cycle of Japonica Rice Under Increased Nitrogen Fertilization. Rice 2023, 16, 34, doi:10.1186/s12284-023-00651-2.
49 Kim, B.; Westerhuis, J.A.; Smilde, A.K.; Floková, K.; Suleiman, A.K.A.; Kuramae, E.E.; Bouwmeester, H.J.; Zancarini, A. Effect of Strigolactones on Recruitment of the Rice Root-Associated Microbiome. FEMS Microbiol Ecol 2022, 98, fiac010 doi:10.1093/femsec/fiac010.
- How frog activities specifically alter water chemistry/microbiota.
Frogs are closely related to the compositional structure of water and soil microorganisms. A previous study has shown that frog-aggregation soil was markedly increased in sulphur respiration and hydrocarbon degradation (Liu et al., 2024). It has also been hypothesized that frog feces and skin carry a large number of microorganisms that enter the water directly or indirectly, so frog behavior affects the microbial composition of the water (Sammon et al., 2010). In addition, we have added it to the manuscript.
Liu S, Imad S, Hussain S, Xiao S, Yu X, Cao H. Sex, health status and habitat alter the community composition and assembly processes of symbiotic bacteria in captive frogs. BMC Microbiology. 2024; 24(1): 34. doi: 10.1186/s12866-023-03150-y.
Sammon NB, Harrower KM, Fabbro LD, Reed RH. Microfungi in drinking water: the role of the frog Litoria caerulea. Int J Environ Res Public Health. 2010 ,7(8),3225-3234, doi: 10.3390/ijerph7083225. Epub 2010 Aug 19..
Similarly, frog activities (e.g., predation, excretion, etc.) could change the physico-chemical properties and biological composition of the water, leading to possible differences between the rice monoculture group and the rice-frog co-cropping group in environmental factors such as dissolved oxygen, temperature, and pH in the water. A previous study has shown that frog-aggregation soil was markedly increased in sulfur respiration and hydrocarbon degradation [26]. It has also been hypothesized that frog feces and skin carry a large number of microorganisms that enter the water column directly or indirectly, so frog behavior affects the microbial composition of the water [16,51]. (Refer to the third paragraph of the section “4.1. Microbe diversity in rice paddy water from different cropping patterns and periods”, L530~539)
Added reference:
26 Liu, S.; Imad, S.; Hussain, S.; Xiao, S.; Yu, X.; Cao, H. Sex, Health Status and Habitat Alter the Community Composition and Assembly Processes of Symbiotic Bacteria in Captive Frogs. BMC Microbiol 2024, 24, 34, doi:10.1186/s12866-023-03150-y.
51 Sammon NB, Harrower KM, Fabbro LD, Reed RH. Microfungi in drinking water: the role of the frog Litoria caerulea. Int J Environ Res Public Health. 2010 ,7,3225-3234, doi: 10.3390/ijerph7083225. Epub 2010 Aug 19.
- The study discusses environmental impacts on microbiota, yet there is no data provided on key water quality parameters. These should be reported and discussed, as they may strongly influence microbial community structure.
Response: We appreciate the reviewer for highlighting this critical point. Water quality parameters (e.g., temperature, pH, dissolved oxygen, N/P concentrations) are indeed core environmental drivers of microbial community structure (Wang et al., 2019; Chen et al., 2020). Due to limitations in field sampling, continuous rainfall during sampling periods made water quality control difficult, thus, we did not collect synchronous water quality data. To address this, we've added this limitation in conclusion to account for this.
Wang S, Wu Q, Nie Y, Wu J, Xu Y. Construction of synthetic microbiota for reproducible flavor compound metabolism in Chinese light-aroma-type liquor produced by solid-state fermentation. Applied and Environmental Microbiology 2019, 85(10): e03090-18, doi: 10.1128/AEM.03090-18.
Chen Y, Sun R, Sun T, Chen P, Yu ZY, et al. Evidence for involvement of keystone fungal taxa in organic phosphorus mineralization in subtropical soil and the impact of labile carbon. Soil Biology and Biochemistry, 2020, 148: 107900, doi: 10.1016/j.soilbio.2020.107900.
However, this study has limitations. For instance, relying solely on 16S rRNA sequencing limits functional insights, and the single-season field trial restricts long-term validity. At the same time, changes in some environmental factors (e.g., temperature, pH) in the field experiment may affect the experiment to some extent. What’s more, pathogenic bacteria detected in samples weren’t experimentally validated for virulence. (Refer to the second paragraph of the section “5. Conclusions”, L819~823)
- The conclusions about “stress resistance” and “pathogenicity” based on BugBase predictions should be discussed with more caution. These predictions are inference-based and should not be overstated without supporting functional validation.
Response: Thanks to your scientific advice, we have added literature support and adjusted the presentation in the original manuscript.
The PICRUSt2 functional prediction difference analysis showed that the functional genes significantly enriched in the rice-frog co-cropping group were mainly involved in amino acid synthesis, metabolism, and antibiotic synthesis, while cell antigens were significantly enriched in the water of and the rice-monocropping group are quite different. This suggests that the rice-frog co-cropping group promoted amino acid synthesis and metabolism in the paddy water, while also inhibiting some harmful bacteria in the paddy water. In addition, this study found that microbes related to elemental movement and stress resistance were significantly enriched in the water of the rice-frog co-cropping group during the embryonic, full heading, and ripening stages of rice. This suggests that the rice-frog co-cropping model may be able to improve the element cycling and stress resistance of paddy ecosystems to a certain extent. These phenomena may be related to the activities and excretion of the black-spotted frog, and have contributed to improved water fertility in paddy fields and the better growth of rice. However, the potential pathogenicity of microbiota in the water of the high-density rice-frog co-cropping group was significantly higher at the embryonic, full heading, and ripening stages of rice, possibly due to the high density of frogs. Additionally, studies in aquaculture have shown that high rearing density can increase potential pathogenicity [56], but no studies have found that the rice-frog co-cropping model improves the resilience of paddy aquatic ecosystems. (Refer to the sixth paragraph of the section “4.1. Microbe diversity in rice paddy water from different cropping patterns and periods”, L612~629)
This suggests that reasonable placement of black-spotted frogs in paddy fields may promote the utilization of nutrients and degradation of harmful substances in paddy fields to a certain extent, as well as improve the stress resistance of black spotted frogs, even in the presence of some pathogenic bacteria. (Refer to the last paragraph of the section “4.2. Gut microbiota diversity of frogs and differences between cropping patterns and periods”, L730~733)
Added reference:
56 Ellison, A.R.; Uren Webster, T.M.; Rodriguez-Barreto, D.; de Leaniz, C.G.; Consuegra, S.; Orozco-terWengel, P.; Cable, J. Comparative Transcriptomics Reveal Conserved Impacts of Rearing Density on Immune Response of Two Important Aquaculture Species. Fish Shellfish Immunol 2020, 104, 192–201, doi:10.1016/j.fsi.2020.05.043.
- The Discussion is lengthy; consider condensing redundant paragraphs.
Response: Thank you for the feedback on streamlining the Discussion section. We have revised the section to enhance conciseness by consolidating redundant content and prioritizing key findings.
In the rice monoculture group, enriched microbes such as Nitrosomonadaceae_unclassified (possibly related to the nitrogen cycling process), interact with the rice root system, making nitrogen sources available for the rice. This interaction may help improve nitrogen availability in the soil and water, promoting nitrogen uptake and thus rice growth. (Refer to the fourth paragraph of the section “4.1. Microbe diversity in rice paddy water from different cropping patterns and periods”, L579~583)
In the low-density rice-frog co-cropping group, the existence of Xenorhabdus may be related to the presence of frogs. At the same time, the predatory behavior of frogs may have influenced the abundance and distribution of other microbes, indirectly creating a more suitable living space for Xenorhabdus. As a symbiotic bacterium of entomopathogenic nematodes, Xenorhabdus may participate in controlling these pests in the rice fields [54,55], and by taking advantage of this, the use of chemical pesticides can be reduced or abandoned, resulting in organic farming systems. Meanwhile, frog predation likely influences microbial communities, creating niches favorable for Xenorhabdus. the rice-frog co-cropping mode itself may reduce the number of pests through predation and other behaviors of frogs, which, combined with the biological control by Xenorhabdus, is expected to establish a more effective biological control system. On the other hand, rice monoculture systems may rely more on chemical pest control, posing environmental risks [14]. due to the characteristics of the microbial community in the rice monoculture group, there may be a greater reliance on chemical control for pests and diseases, which may negatively impact the environment and the quality of the agricultural products. However, if Xenorhabdus adversely affects frogs or other beneficial organisms, it may disrupt the balance of the ecosystem and indirectly affect the rice-growing environment, which is something that should be explored in the future. Thus, future studies should assess whether Xenorhabdus impacts frog health or ecosystem balance. (Refer to the fifth paragraph of the section “4.1. Microbe diversity in rice paddy water from different cropping patterns and periods”, L593~611)
The PICRUSt2 functional prediction difference analysis showed that the functional genes significantly enriched in the rice-frog co-cropping group were mainly involved in amino acid synthesis, metabolism, and antibiotic synthesis, while cell antigens were significantly enriched in the water of and the rice-monocropping group are quite different. (Refer to the sixth paragraph of the section “4.1. Microbe diversity in rice paddy water from different cropping patterns and periods”, L612~615)
Gut microbes are closely linked to host pathogen defense associated with the host’s defense against pathogens [19]. At the phylum level, The Proteobacteria and Firmicutes were the most abundant microbes in the frog’s gut, which is consistent with the results of previous studies [57,58]. Some microbes in the Firmicutes help the host to digest and absorb nutrients, while the Proteobacteria are metabolically diverse and can adapt to a wide range of environmental conditions [45]. The increase in Proteobacteria enrichment in the frog’s gut of the black-spotted frog during the full heading stage of rice may result from dietary introduction of pollen or plant secretions, potentially disrupting gut microbiota homeostasis be due to the introduction of substances (such as pollen or plant secretions) into the food chain or living environment, which led to the disruption of the homeostasis of the gut microbiota [59,60]. (Refer to the first paragraph of the section “4.2. Gut microbiota diversity of frogs and differences between cropping patterns and periods”, L638~648)
Lactococcus and Cetobacterium are the resident bacteria in the gut of the black-spotted frog, easily adapting to the basic environmental and nutritional conditions of the gut. Moreover, changes in the relative abundance of Pseudoalteromonas and Vibrio may be related to the carriage of microbiota in feed and the influence of environmental microbiota. These two microbial groups were more abundant in the frog guts of the low-density group at the embryonic and full heading stages of rice, which may be due to the range of activity and food of the black-spotted frogs in the low-density group being different from that of the high-density group. (Refer to the second paragraph of the section “4.2. Gut microbiota diversity of frogs and differences between cropping patterns and periods”, L661~668)
The α- and β-diversity indices of the gut microbiota of frogs were unaffected not significantly influenced by the placement frog density of frogs. However, there were but shown significant differentces in the structure of the gut microbiota during across different periods of rice growth. A higher α-diversity index implies a richer variety of microbes in the gut of frogs and a relatively balanced distribution of each taxon. A higher α-diversity index indicated greater microbial richness and evenness in frog guts. (Refer to the third paragraph of the section “4.2. Gut microbiota diversity of frogs and differences between cropping patterns and periods”, L678~683)
The occurrence of more endemic species in the gut of black-spotted frogs In the high-density rice-frog co-cropping group, frog guts harbored more endemic species, likely due to the model’s complex ecological network fostering tighter symbiotic relationships. may be that this model provides a more complex and diverse ecological environment. Species interactions are more frequent and complex in high-density rice-frog co-cropping environments, and the rice, frogs, and other organisms form a tighter ecological network with more symbiotic, mutualistic, and other relationships. It is valuable to note that Sphingopyxis can survive in a wide range of extreme environments and degrade exogenous substances and environmental pollutants, with the high-density rice-frog symbiosis group potentially providing the right environmental conditions suitable for the growth of Sphingopyxis. This resulted in the presence of Sphingopyxis in the intestinal tract of the black-spotted frog. (Refer to the fourth paragraph of the section “4.2. Gut microbiota diversity of frogs and differences between cropping patterns and periods”, L693~704)
The English could be improved to more clearly express the research.
Response: Thank you for your interest in the linguistic aspects of our articles, and we understand that clear and concise communication is essential for scientific writing. Therefore, we have touched up the article in the native language before submission (proof is provided), and the revised parts have been double-checked with the help of software such as “Grammarly”.
Thank you again for your insightful comments and rigorous scrutiny of our manuscript. We sincerely appreciate the time and effort you have invested in reviewing our work. We hope the revised version meets your expectations, and we look forward to your further feedback.
Yours sincerely,
Zhangyan Zhu

Reviewer 2 Report
Comments and Suggestions for Authors
This is an interesting study. However, here are some of my comments to improve the manuscript
- Including soil microbiota would have provided a more comprehensive view of the rice–frog–environment interaction.
- While pooling improves cost efficiency, it may obscure individual-level microbiota variation. If sequencing depth allows, separate sequencing could improve resolution.
- No mention of monitoring water temperature, pH, dissolved oxygen, or nutrient levels (e.g., nitrogen, phosphorus), which could significantly influence microbial communities.
- Details about DNA extraction kits, and downstream analysis tools (e.g., QIIME2, DADA2) are not included here but are essential for reproducibility.
Comments on the Quality of English Language
Overall it good
Author Response
This is an interesting study. However, here are some of my comments to improve the manuscript.
Response: We greatly appreciate your positive feedback and thoughtful comments, which have significantly helped us improve the manuscript.
Including soil microbiota would have provided a more comprehensive view of the rice–frog–environment interaction.
Response: Thank you very much for your suggestion, which is a very interesting research element. However, shortly before this, there was already a paper on the effect of rice-frog working pattern on soil physicochemical properties and soil microorganisms (Ma et al., 2024), which explained in detail the changes in the soil microbiota under the rice-frog co-cropping pattern. So we decided to focus on the relationship between rice paddy water and frog gut microorganisms. If there is a subsequent opportunity, we will also consider a joint analysis of the microbiota of soil-paddy water-frog gut to explore the interaction between rice-frog-environment more comprehensively and rationally.
Ma, Y.; Yu, A.; Zhang, L.; Zheng, R. Effects of Rice-Frog Co-Cropping on the Soil Microbial Community Structure in Reclaimed Paddy Fields. Biology (Basel) 2024, 13, 396, doi:10.3390/biology13060396.
While pooling improves cost efficiency, it may obscure individual-level microbiota variation. If sequencing depth allows, separate sequencing could improve resolution.
Response: Thank you for making this important point about the impact of pooled sequencing on microbial variation at the individual level. We fully recognize the trade-off between cost-effectiveness and resolution of our approach, which may be able to be applied to our subsequent experiments.
No mention of monitoring water temperature, pH, dissolved oxygen, or nutrient levels (e.g., nitrogen, phosphorus), which could significantly influence microbial communities.
Response: We appreciate the reviewer for highlighting this important point. This study focused on the structural characteristics and interaction mechanisms of microbial communities in the rice-frog symbiotic system. The environmental factors such as water temperature, pH, dissolved oxygen, and nutrient salts (N and P) were not synchronized because of the limitations of the sample collection scale and experimental conditions, as well as the fact that the experimental plots were adjacent to each other and were located in generally similar environmental conditions. To address this gap, we have included the limitations of this study in the conclusion.
However, this study has limitations. For instance, relying solely on 16S rRNA sequencing limits functional insights, and the single-season field trial restricts long-term validity. At the same time, changes in some environmental factors (e.g., temperature, pH) in the field experiment may affect the experiment to some extent. What’s more, pathogenic bacteria detected in samples weren’t experimentally validated for virulence. (Refer to the second paragraph of the section “5. Conclusions”, L819~823)
Details about DNA extraction kits, and downstream analysis tools (e.g., QIIME2, DADA2) are not included here but are essential for reproducibility.
Response: We do have a poor description here, thank you very much for pointing this out and we have amended the content.
- The DNA Extraction Kit
The information about the DNA extraction kit is described in the original manuscript, the details are as follows.
The TGuide S96 Magnetic Bead kit for soil/fecal genomic DNA extraction (Tiangen Biochemical Science and Technology (Beijing) Co., Ltd., DP812) was used to extract nucleic acids from the water and gut samples. (Refer to the first paragraph of the section “2.3. DNA extraction and high-throughput sequencing analysis”, L146~148)
- Downstream analysis tools
For the detailed information and processes of the downstream tools for microbial analysis, we did not originally describe them, so a new section has been added to describe them in detail.
2.4. Sequencing analysis and microbial taxonomic identification
The microbiome was analyzed using QIIME2 (version 2021. 8) [31]. Raw sequence data were decoded using the demux plugin, and primers were excised using the cutadapt plugin [32]. The sequences were processed for quality filtering, denoising, splicing, and chimera removal using the DADA2 plugin [33] . Sequences were then merged at 100% sequence similarity to generate characteristic sequence amplicon sequence variation (ASV) and abundance data tables. The ASV feature sequences were compared with reference sequences in the database using the Greengenes database to obtain taxonomic information corresponding to each ASV. ASVs with abundance values below 0.001% of the total number of sequenced samples were excluded, and then the abundance matrix of the excluded rare ASVs was analyzed. (Refer to the section “2.4. Sequencing analysis and microbial taxonomic identification”, L159~170)
Added reference
31 Bolyen, E.; Rideout, J.R.; Dillon, M.R.; Bokulich, N.A.; Abnet, C.C.; Al-Ghalith, G.A.; Alexander, H.; Alm, E.J.; Arumugam, M.; Asnicar, F.; et al. Reproducible, Interactive, Scalable and Extensible Microbiome Data Science Using QIIME 2. Nat Biotechnol 2019, 37, 852–857, doi:10.1038/s41587-019-0209-9.
32 Martin, M. Cutadapt Removes Adapter Sequences from High-Throughput Sequencing Reads. EMBnet J 2011, 17, 10, doi:10.14806/ej.17.1.200.
33 Chen, J.; Gu, W.; Tao, J.; Xu, Y.; Wang, Y.; Gu, J.; Du, S. The Effects of Organic Residue Quality on Growth and Reproduction of Aporrectodea Trapezoides under Different Moisture Conditions in a Salt-Affected Agricultural Soil. Biol Fertil Soils 2017, 53, 103–113, doi:10.1007/s00374-016-1158-9.
The English could be improved to more clearly express the research.
Response: Thank you for your interest in the linguistic aspects of our articles, and we understand that clear and concise communication is essential for scientific writing. Therefore, we have touched up the article in the native language before submission (proof is provided), and the revised parts have been double-checked with the help of software such as “Grammarly”.
Thank you again for your insightful comments and rigorous scrutiny of our manuscript. We sincerely appreciate the time and effort you have invested in reviewing our work. We hope the revised version meets your expectations, and we look forward to your further feedback.
Yours sincerely,
Zhangyan Zhu

Reviewer 3 Report
Comments and Suggestions for Authors
REVIEW FOR THE MANUSCRIPT
JOURNAL: Microorganisms (ISSN 2076-2607)
Manuscript ID: microorganisms-3698104
Type: Original Article
Title: Frog culturing density and growth stage affect both paddy field and frog gut microbial communities in rice-frog co-cropping models
Section: Gut Microbiota
OVERALL COMMENTS
- In this manuscript, the authors intended to investigate the modifications in the microbiotal community structure of paddy field water and the frog’s gut in rice-frog co-cropping models at different rice growth stages. Their results showed that Proteobacteria were the dominant taxa in paddy field water, while the most abundant phyla in the guts of frogs were Firmicutes and Actinobacteriota. The α-diversity indices of the microbiotas in the paddy field water and the frog’s guts were not significantly affected by the density of frogs. In summary, they suggest that the microbial community structure of paddy field water and frog guts in the rice-frog co-cropping model is correlated and influenced by the period of rice growth, while the increased prevalence of Xenorhabdus in the low-density group promoted the growth of entomopathogenic nematodes and controlled the number of pests in the rice field. In addition, the frog’s guts of the low-density group were markedly at resisting external disturbances. This provides a scientific reference for adjusting frog culture management measures according to changes in the rice’s growth stage and main-taining the balance and stability of the water and frog gut microbiotas.
TITLE
The title of the manuscript is too long. Is there a possibility of reducing it and making it clearer?
ABSTRACT
This section seems not to be adequate regarding the number of words. Please check MDPI guidelines.
KEYWORDS
The keywords are adequate.
INTRODUCTION
Please include newer references in this section. Are there references published in 2024 and 2025? Furthermore, I believe that the number of studies in this section should be increased.
- METHODS
In this section, it is necessary to mention variables that could deeply interfere with the results (such as temperature, pH, etc).
What were the frog management techniques? Did the authors follow any international criteria related to animal management?
Please describe the approval by an Ethics Committee. I could not find it in the Methods section, and not in:
Institutional Review Board Statement: ?
Informed Consent Statement: ?
- RESULTS
This section is adequate, and the figures are adequate. However, they are too much. Is it possible to reduce some?
Figures 6 and 12 are too small. I suggest using a, b,c, … not side by side.
- DISCUSSION
This section is too long. Many parts have already been mentioned in the Introduction and Results sections.
Another example: Is the possible ecological effects of detected microorganisms (e.g., Sphingopyxis) necessary? Or is it a speculation?
Are there new studies that can be used to compare the results?
- CONCLUSIONS
This section also repeats information described in the Results and Discussion.
- LIMITATIONS AND FUTURE PERSPECTIVES
Please include the limitations and the future perspectives for this study.
As an example of limitation: The lack of an approach to these environmental factors may affect the robustness of the study. It is necessary to comment on this.
- REFERENCES
It is necessary to include newer references mainly in the Introduction section.
I also miss references mainly related to international literature, showing whether or not there is a wide application of the observed results.
Author Response
In this manuscript, the authors intended to investigate the modifications in the microbiotal community structure of paddy field water and the frog’s gut in rice-frog co-cropping models at different rice growth stages. Their results showed that Proteobacteria were the dominant taxa in paddy field water, while the most abundant phyla in the guts of frogs were Firmicutes and Actinobacteriota. The α-diversity indices of the microbiotas in the paddy field water and the frog’s guts were not significantly affected by the density of frogs. In summary, they suggest that the microbial community structure of paddy field water and frog guts in the rice-frog co-cropping model is correlated and influenced by the period of rice growth, while the increased prevalence of Xenorhabdus in the low-density group promoted the growth of entomopathogenic nematodes and controlled the number of pests in the rice field. In addition, the frog’s guts of the low-density group were markedly at resisting external disturbances. This provides a scientific reference for adjusting frog culture management measures according to changes in the rice’s growth stage and maintaining the balance and stability of the water and frog gut microbiotas.
Response: Thank you for your thorough summary of our manuscript. We appreciate your careful reading and thoughtful feedback, which confirms that the core findings and scientific significance of our study have been effectively communicated.
TITLE
The title of the manuscript is too long. Is there a possibility of reducing it and making it clearer?
Response: Thank you very much for your suggestion, we have removed the repetitive and unnecessary this from the title and have decided to streamline the title:
Frog density and growth stage of rice impact paddy field and gut microbial communities in rice-frog co-cropping model. (Refer to the title, L2~4)
ABSTRACT
This section Refer toms not to be adequate regarding the number of words. Please check MDPI guidelines.
Response: Thank you for pointing out this problem with the abstract, which we have revised as requested. Repetitive and excessive results have been removed from the abstract to make it short and easy to read.
Modified abstract:
The black-spotted frog (Pelophylax nigromaculatus) is a common economic species in the rice-frog ecological cropping mode. The present study investigated microbial community structures in paddy water and black-spotted frog’s guts across rice monoculture and low/high-density rice-frog co-cropping systems at four rice growth stages. Proteobacteria dominate in paddy water, while frog guts are enriched in Firmicutes and Actinobacteriota. Frog density shows no impact on α-diversity, but rice growth stages significantly alter Shannon, Simpson, and Pielou_e indices (p < 0.01). Co-cropping promotes amino acid synthesis, elemental cycling, and stress tolerance in paddy water microbiota, which is more diverse than gut microbiota. Strong correlations exist between paddy water and gut microbiotas, with Limnohabitans linked to gut diversity (p < 0.05). Low-density co-cropping enhances Xenorhabdus for pest control and stabilizes gut microbiota, offering insights for managing rice-frog systems based on rice growth stages. (Refer to the abstract, L11~45)
KEYWORDS
The keywords are adequate.
Response: Thank you for acknowledging the adequacy of the keywords. We appreciate your confirmation and will ensure they accurately reflect the study's focus in the final manuscript.
INTRODUCTION
Please include newer references in this section. Are there references published in 2024 and 2025? Furthermore, I believe that the number of studies in this section should be increased.
Response: We sincerely appreciate your valuable suggestion regarding the references in our manuscript. We fully recognize the significance of incorporating up-to-date literature to enhance the currency and comprehensiveness of our study.
To address this, we have conducted an extensive search across multiple databases, specifically focusing on literature published in 2024 and 2025. As a result, we have identified several highly relevant studies that have been integrated into the manuscript. In addition to adding recent references, we have also substantially increased the number of studies cited throughout the manuscript. We have combed through the existing literature more comprehensively, incorporating works from different research groups and geographical regions to strengthen the foundation of our arguments. This has led to a more diverse and well-rounded view of the topic, covering a broader range of experimental approaches and theoretical frameworks.
All the new references have been carefully integrated into the text, with appropriate in-text citations and corresponding entries in the reference list, following the journal's formatting guidelines. We believe these revisions will significantly enhance the quality and credibility of our work, and we are grateful for your guidance in this regard.
Added references published in 2024 and 2025:
21 Khaledi, M.; Poureslamfar, B.; Alsaab, H.O.; Tafaghodi, S.; Hjazi, A.; Singh, R.; Alawadi, A.H.; Alsaalamy, A.; Qasim, Q.A.; Sameni, F. The Role of Gut Microbiota in Human Metabolism and Inflammatory Diseases: A Focus on Elderly Individuals. Ann Microbiol 2024, 74, 1, doi:10.1186/s13213-023-01744-5.
27 Dang, W.; Zhang, J.-H.; Cao, Z.C.; Yang, J.-M.; Lu, H.L. Environmentally Relevant Levels of Antiepileptic Carbamazepine Altered Intestinal Microbial Composition and Metabolites in Amphibian Larvae. Int J Mol Sci 2024, 25, 6950, doi:10.3390/ijms25136950.
37 Yang, L.; Pan, R.; Wang, S.; Zhu, Z.; Li, H.; Yang, R.; Sun, X.; Ge, B. Macrofaunal Biodiversity and Trophic Structure Varied in Response to Changing Environmental Properties along the Spartina Alterniflora Invasion Stages. Mar Pollut Bull 2025, 214, 117756, doi:10.1016/j.marpolbul.2025.117756.
47 Zhou, H.Z.; Wang, B.Q.; Ma, Y.H.; Sun, Y.Y.; Zhou, H.L.; Song, Z.; Zhao, Y.; Chen, W.; Min, J.; Li, J.W.; et al. The Combination of Metagenomics and Metabolomics Reveals the Effect of Nitrogen Fertilizer Application Driving the Remobilization of Immobilization Remediation Cadmium and Rhizosphere Microbial Succession in Rice. J Hazard Mater 2025, 487, 137117, doi:10.1016/j.jhazmat.2025.137117.
METHODS
In this section, it is necessary to mention variables that could deeply interfere with the results (such as temperature, pH, etc).
Response: We appreciate the reviewer for highlighting this important point. This study focused on the structural characteristics and interaction mechanisms of microbial communities in the rice-frog symbiotic system. The environmental factors such as water temperature, pH, dissolved oxygen, and nutrient salts (N and P) were not synchronized because of the limitations of the sample collection scale and experimental conditions, as well as the fact that the experimental plots were adjacent to each other and were located in generally similar environmental conditions. To address this gap, we plan to incorporate monitoring of environmental factors in similar studies, while data from this study may provide a baseline reference for microbial communities in this direction.
What were the frog management techniques? Did the authors follow any international criteria related to animal management?
Response: Thank you for raising this important question regarding frog management practices and compliance with international animal care standards. We understand the critical need to ensure ethical treatment of animals in research, and we have addressed this by providing the following clarifications.
- Detailed Description of Frog Management Techniques
In the revised “Methods and materials” section, we have added the following.
Jindadi feed (Zhejiang Jindadi Bio-technology Co., Ltd., Shaoxing, China) was used as the feed for the frogs in this study. The feeding amount was approximately 5% of the frog’s weight. The frog feed contained ≥ 40% crude protein, ≥ 4.5% crude fat, ≤ 8% crude fiber, ≤ 18% crude ash, ≥ 1% total phosphorus, and ≥ 1.8% lysine. After being put in the paddy field, the frogs were fed at 17:00 every day. (Refer to the second paragraph of the section “2.2. Sample collection and processing”, L140~144)
- Compliance with International Animal Care Standards
All frog handling and experimental procedures were conducted following the Guide for the Care and Use of Laboratory Animals (8th edition, National Academies Press, 2011) and approved by the Animal Care and Use Committee (IACUC) of Zhejiang Normal University, protocol number ZSDW2022055."
Please describe the approval by an Ethics Committee. I could not find it in the Methods section, and not in:
Institutional Review Board Statement: ?
Informed Consent Statement: ?
Response: We apologize for the omission of this important information, which has now been added.
Institutional Review Board Statement: All experimental procedures involving frogs were approved by the Institutional Review Board of Zhejiang Normal University (ZSDW2022055). (L871~872)
Meanwhile, we have also prepared and attached both Chinese and English versions of the materials related to the ethical review.
RESULTS
This section is adequate, and the figures are adequate. However, they are too much. Is it possible to reduce some?
Response: Thank you for your feedback on the results. We understand your concern that too many results may overwhelm readers, and we appreciate the opportunity to optimize the presentation of results. To address this issue, we have carefully reviewed the results section and streamlined some sections that were too lengthy.
Previous the second paragraph of “3.1.4. Predictive analysis of water microbiota”, L309~323
BugBase phenotypic predictions (Figure 6) showed that the relative abundance of ‘anaerobic’, ‘contains mobile elements’, ‘facultatively anaerobic’, ‘gram positive’, ‘potentially pathogenic’, and ‘stress tolerant’ genes was higher in the paddy field water of the rice-frog co-cropping group compared to the rice monoculture group, and that the relative abundance of potentially pathogenic genes was higher in the paddy field water of the high-density rice-frog co-cropping group compared to the low-density rice-frog co-cropping group. The anaerobic bacteria in the rice-frog co-cropping group were significantly more than those in the rice monoculture group at the tillering, embryonic, and full heading stages of rice, and their relative abundance was significantly lower at the ripening stage. Except for the tillering stage, the ‘contains mobile elements’ abundance in the rice-frog co-cropping group was significantly higher than that of the rice monoculture group. The gram-positive bacteria in the rice-frog co-cropping group were significantly more than those in the rice monoculture group at the tillering and ripening stages. The high-density rice-frog co-cropping group had more potentially pathogenic microbes than the rice monoculture group and the low-density rice-frog co-cropping group at the tillering, embryonic, and full heading stages. The ‘stress tol-erance’ genes in the paddy field water microbiota in the rice frog co-cropping group were significantly more compared with the rice monoculture group at the embryonic, full heading, and ripening stages of rice.
Modified the second paragraph of “3.1.4. Predictive analysis of water microbiota”, L297~309
BugBase phenotypic predictions (Figure 6) showed that paddy field water in the rice-frog co-cropping group had higher relative abundances of ‘anaerobic’, ‘contains mobile elements’, ‘facultatively anaerobic’, ‘gram positive’, ‘potentially pathogenic’, and ‘stress tolerant’ genes compared to the rice monoculture group. Firstly, ‘Anaerobic’ and ‘stress-tolerant’ gene abundances were significantly higher in the co-cropping group during the rice tillering, embryonic, and full heading stages, but lower at ripening (except for’ stress-tolerant’ genes, which remained high). Secondly, ‘contains mobile elements’ gene abundance was higher in co-cropping groups at all stages except tillering. Thirdly, gram-positive bacteria were more abundant in co-cropping groups during tillering and ripening. Lastly, high-density co-cropping groups showed elevated potentially pathogenic microbes compared to low-density co-cropping and monoculture groups at tillering, embryonic, and full heading stages.
Previous the first paragraph of “3.2.4. Predictive analysis of gut microbial function”, L402~419
Differential analysis of STAMP using the PICRUSt2 function prediction showed that 22 differentially KO-enriched functional genes were identified on the KEGG pathway ‘Level 3 metabolic pathways’ at the ripening stage of rice (Figure 11). The functional pathways of the 12 KO-represented genes were significantly enriched in the low-density rice-frog co-cropping group, including ‘two-component system’, ‘African trypanosomiasis’, ‘huntington's disease’, ‘penicillin and cephalosporin biosyn-thesis’, ‘glutathione metabolism’, ‘bladder cancer’, ‘ethylbenzene degradation’, ‘protein kinases’, ‘alpha-Linolenic acid metabolism’, ‘alzheimer's disease’, ‘parkinson's disease’, and ‘atrazine degradation’, which were significantly more than those in the high-density rice-frog co-cropping group (p < 0.05). At the ripening stage of rice, the expression of ‘cysteine and methionine metabolism’, ‘RNA polymerase’, ‘protein processing in endoplasmic reticulum’, ‘alanine, aspartate, and glutamate metabolism’, ‘ribosome’, ‘zeatin biosynthesis’, ‘lysine biosynthesis’, ‘amino acid related enzymes’, ‘aminoacyl-tRNA biosynthesis’, and ‘polyketide sugar unit biosynthesis’ genes was significantly higher in frog’s gut samples from the high-density rice-frog co-cropping group than from the low-density rice-frog co-cropping group (p < 0.05).
Modified the first paragraph of “3.2.4. Predictive analysis of gut microbial function”, L419~426
STAMP differential analysis via PICRUSt2 functional prediction identified 22 differentially enriched KO genes in KEGG Level 3 metabolic pathways at rice ripening stage (Figure 11). On one hand, 12 KO-represented pathways were significantly enriched in low-density co-cropping groups, including ‘two-component system’, ‘glutathione metabolism’, ‘atrazine degradation’, and neurological disease-related pathways (e.g., Huntington's, Alzheimer's, Parkinson's disease), with abundances higher than high-density groups (p < 0.05). On the other hand, high-density co-cropping groups showed elevated expression of cysteine/methionine metabolism, ribosome, zeatin biosynthesis, and amino ac-id-related pathways (e.g., ‘aminoacyl-tRNA biosynthesis’) in frog gut samples com-pared to low-density groups (p < 0.05).
Previous the second paragraph of “3.2.4. Predictive analysis of gut microbial function”, L427~442
Based on the BugBase phenotypic predictions (Figure 12), the relative abundance of anaerobic bacteria was elevated in the guts of the high-density rice-frog co-cropping group, compared to the low-density rice-frog co-cropping group. At the tillering, embryonic, and ripening stages of rice, the relative abundance of ‘contains mobile element’ genes of microbiota in the gut of frogs in the low-density rice-frog co-cropping group was higher, while the relative abundance of facultatively anaerobic genes was lower compared to that of the high-density rice-frog co-cropping group. The relative abundance of the ‘potentially pathogenic’ and ‘stress tolerance’ genes was higher in the low-density rice-frog co-cropping group at the tillering, full heading, and ripening stages of rice compared to the high-density rice-frog co-cropping group. The relative abundance of gram-positive bacteria was higher in the low-density rice-frog co-cropping group at the tillering and embryonic stages of rice. In comparison, the relative abundance of gram-negative and biofilm-forming-associated bacteria was lower than the high-density rice-frog co-cropping group. However, the relative abundance of gram-positive, gram-negative, and biofilm-forming bacteria was higher in the high-density rice-frog co-cropping group at the full heading and ripening stages of rice.
Modified the second paragraph of “3.2.4. Predictive analysis of gut microbial function”, L442~450
BugBase phenotypic analysis (Figure 12) showed higher anaerobic bacterial abundance in high-density co-cropping frog guts versus low-density groups. Firstly, low-density groups had elevated ‘contains mobile element’ genes at tillering, embryonic, and ripening stages, but lower ‘facultative anaerobic’ genes compared to high-density groups. Secondly, ‘potentially pathogenic’ and ‘stress tolerant’ genes were more abundant in low-density groups at tillering, full heading, and ripening stages. Finally, gram-positive bacteria dominated low-density guts at tillering and embryonic stages, while high-density groups showed higher gram-negative and biofilm-forming bacteria at full heading and ripening.
Figures 6 and 12 are too small. I suggest using a, b,c, … not side by side.
Response: Thank you for your feedback on the figure clarity. We agree that the current layout of Figures 6 and 12 compromises visibility, and we have revised them as follows to enhance readability.
DISCUSSION
This section is too long. Many parts have already been mentioned in the Introduction and Results sections.
Response: Thank you for highlighting the need to streamline this section. We have rigorously edited the content to eliminate redundancy with the Introduction and Results sections, focusing solely on interpretive analysis.
In the rice monoculture group, enriched microbes such as Nitrosomonadaceae_unclassified (possibly related to the nitrogen cycling process), interact with the rice root system, making nitrogen sources available for the rice. This interaction may help improve nitrogen availability in the soil and water, promoting nitrogen uptake and thus rice growth. (Refer to the fourth paragraph of the section “4.1. Microbe diversity in rice paddy water from different cropping patterns and periods”, L579~583)
The PICRUSt2 functional prediction difference analysis showed that the functional genes significantly enriched in the rice-frog co-cropping group were mainly involved in amino acid synthesis, metabolism, and antibiotic synthesis, while cell antigens were significantly enriched in the water of and the rice-monocropping group are quite different. (Refer to the sixth paragraph of the section “4.1. Microbe diversity in rice paddy water from different cropping patterns and periods”, L612~615)
The α- and β-diversity indices of the gut microbiota of frogs were unaffected not significantly influenced by the placement frog density of frogs. However, there were but shown significant differentces in the structure of the gut microbiota during across different periods of rice growth. A higher α-diversity index implies a richer variety of microbes in the gut of frogs and a relatively balanced distribution of each taxon. A higher α-diversity index indicated greater microbial richness and evenness in frog guts. (Refer to the third paragraph of the section “4.2. Gut microbiota diversity of frogs and differences between cropping patterns and periods”, L678~683)
Another example: Is the possible ecological effects of detected microorganisms (e.g., Sphingopyxis) necessary? Or is it a speculation?
Response: Thank you for raising this critical point regarding the ecological effects of Sphingopyxis. We acknowledge that clarifying the distinction between observed data and speculative interpretations is essential for scientific rigor, so we made the following adjustments.
In the low-density rice-frog co-cropping group, the existence of Xenorhabdus may be related to the presence of frogs. At the same time, the predatory behavior of frogs may have influenced the abundance and distribution of other microbes, indirectly creating a more suitable living space for Xenorhabdus. As a symbiotic bacterium of entomopathogenic nematodes, Xenorhabdus may participate in controlling these pests in the rice fields [54,55], and by taking advantage of this, the use of chemical pesticides can be reduced or abandoned, resulting in organic farming systems. Meanwhile, frog predation likely influences microbial communities, creating niches favorable for Xenorhabdus. the rice-frog co-cropping mode itself may reduce the number of pests through predation and other behaviors of frogs, which, combined with the biological control by Xenorhabdus, is expected to establish a more effective biological control system. On the other hand, rice monoculture systems may rely more on chemical pest control, posing environmental risks [14]. due to the characteristics of the microbial community in the rice monoculture group, there may be a greater reliance on chemical control for pests and diseases, which may negatively impact the environment and the quality of the agricultural products. However, if Xenorhabdus adversely affects frogs or other beneficial organisms, it may disrupt the balance of the ecosystem and indirectly affect the rice-growing environment, which is something that should be explored in the future. Thus, future studies should assess whether Xenorhabdus impacts frog health or ecosystem balance. (Refer to the fifth paragraph of the section “4.1. Microbe diversity in rice paddy water from different cropping patterns and periods”, L593~611)
Lactococcus and Cetobacterium are the resident bacteria in the gut of the black-spotted frog, easily adapting to the basic environmental and nutritional conditions of the gut. Moreover, changes in the relative abundance of Pseudoalteromonas and Vibrio may be related to the carriage of microbiota in feed and the influence of environmental microbiota. These two microbial groups were more abundant in the frog guts of the low-density group at the embryonic and full heading stages of rice, which may be due to the range of activity and food of the black-spotted frogs in the low-density group being different from that of the high-density group. (Refer to the second paragraph of the section “4.2. Gut microbiota diversity of frogs and differences between cropping patterns and periods”, L661~668)
The occurrence of more endemic species in the gut of black-spotted frogs In the high-density rice-frog co-cropping group, frog guts harbored more endemic species, likely due to the model’s complex ecological network fostering tighter symbiotic relationships. may be that this model provides a more complex and diverse ecological environment. Species interactions are more frequent and complex in high-density rice-frog co-cropping environments, and the rice, frogs, and other organisms form a tighter ecological network with more symbiotic, mutualistic, and other relationships. It is valuable to note that Sphingopyxis can survive in a wide range of extreme environments and degrade exogenous substances and environmental pollutants, with the high-density rice-frog symbiosis group potentially providing the right environmental conditions suitable for the growth of Sphingopyxis. This resulted in the presence of Sphingopyxis in the intestinal tract of the black-spotted frog. (Refer to the fourth paragraph of the section “4.2. Gut microbiota diversity of frogs and differences between cropping patterns and periods”, L693~704)
Are there new studies that can be used to compare the results?
Response: Thank you for your suggestions regarding references, we likewise find the newer references more convincing, so we have updated some of the references. Also considering the publication date of the references and their relevance to this study, we have added the following references.
47 Zhou, H.Z.; Wang, B.Q.; Ma, Y.H.; Sun, Y.Y.; Zhou, H.L.; Song, Z.; Zhao, Y.; Chen, W.; Min, J.; Li, J.W.; et al. The Combination of Metagenomics and Metabolomics Reveals the Effect of Nitrogen Fertilizer Application Driving the Remobilization of Immobilization Remediation Cadmium and Rhizosphere Microbial Succession in Rice. J Hazard Mater 2025, 487, 137117, doi:10.1016/j.jhazmat.2025.137117.
48 Dong, H.; Sun, H.; Chen, C.; Zhang, M.; Ma, D. Compositional Shifts and Assembly in Rhizosphere-Associated Fungal Microbiota Throughout the Life Cycle of Japonica Rice Under Increased Nitrogen Fertilization. Rice 2023, 16, 34, doi:10.1186/s12284-023-00651-2.
49 Kim, B.; Westerhuis, J.A.; Smilde, A.K.; Floková, K.; Suleiman, A.K.A.; Kuramae, E.E.; Bouwmeester, H.J.; Zancarini, A. Effect of Strigolactones on Recruitment of the Rice Root-Associated Microbiome. FEMS Microbiol Ecol 2022, 98, fiac010 doi:10.1093/femsec/fiac010.
56 Ellison, A.R.; Uren Webster, T.M.; Rodriguez-Barreto, D.; de Leaniz, C.G.; Consuegra, S.; Orozco-terWengel, P.; Cable, J. Comparative Transcriptomics Reveal Conserved Impacts of Rearing Density on Immune Response of Two Important Aquaculture Species. Fish Shellfish Immunol 2020, 104, 192–201, doi:10.1016/j.fsi.2020.05.043.
CONCLUSIONS
This section also repeats information described in the Results and Discussion.
Response: Thank you for pointing out the redundancy in this section. We have thoroughly revised the text to streamline the Conclusion section and eliminated overlap with the Results and Discussion sections.
Previous expressions: the first paragraph of the section “5. Conclusions”, L801~825
In this study, the relationship between the gut microbiota of the black-spotted frog and the microbial structure of the paddy field water in different rice-frog co-cropping models was analyzed with 16S high-throughput sequencing. Neither the microbial diversity of the black-spotted frog gut nor the paddy field water was affected by the density of black-spotted frogs, but was, however, affected by different rice growth stages, with a significantly higher diversity in the later stages of rice growth. Compared with the frog’s gut, the microbial structure in the water was richer, with fixed dominant taxa and shared microbes in both, and specific differences. There were significant correlations between the genus Limnohabitans, and the Shannon and Simpson indices of the black-spotted frog gut microbiota. Although the density of frog placement did not affect the diversity of paddy field water and frog gut microbiotas, the enrichment of Xenorhabdus in the low-density group promoted the growth of entomopathogenic nematodes and controlled the number of rice paddy pests. In addition, the frog’s gut microbiota in the low-density group was more stable. Finally, a small number of pathogenic bacteria were found in the paddy field water and the guts of frogs in this study, although they have not been proven pathogenic to black-spotted frogs.
Modified expressions: the first paragraph of the section “5. Conclusions”, L801~825
In this study, the gut microbiota of black-spotted frogs and the microbial structure of paddy water in rice-frog co-cropping models were analyzed with 16S high-throughput sequencing. It was rice growth stage, not frog density, that influenced microbial diversity in both gut and water, with higher diversity in late growth stages. Paddy water harbored richer microbial communities than frog guts, with shared dominant taxa (e.g., Limnohabitans) and specific niche variations. It's worth noting that low-density co-cropping promoted Xenorhabdus enrichment, potentially enhancing entomopathogenic nematode activity for pest control, while showing more stable gut microbiota. Finally, a small number of pathogenic bacteria were found in the paddy field water and the guts of frogs in this study, although they have not been proven pathogenic to black-spotted frogs.
LIMITATIONS AND FUTURE PERSPECTIVES
Please include the limitations and the future perspectives for this study.
Response: Thank you for your advice on limitations and outlook, which is important for our original manuscript. We have added and improved that section as you mentioned.
However, this study has limitations. For instance, relying solely on 16S rRNA sequencing limits functional insights, and the single-season field trial restricts long-term validity. At the same time, changes in some environmental factors (e.g., temperature, pH) in the field experiment may affect the experiment to some extent. What’s more, pathogenic bacteria detected in samples weren’t experimentally validated for virulence. Thus, future studies should employ metagenomics to explore microbial functions, conduct multi-seasonal trials, and validate pathogen impacts through infection assays. This will enhance disease control and advance sustainable rice-frog co-cropping. (Refer to the second paragraph of the section “5. Conclusions”, L826~833)
As an example of limitation: The lack of an approach to these environmental factors may affect the robustness of the study. It is necessary to comment on this.
Response: Thank you for highlighting the importance of addressing study limitations. We have incorporated the following points to acknowledge constraints and outline future directions, ensuring transparency and scientific rigor.
However, this study has limitations. For instance, relying solely on 16S rRNA sequencing limits functional insights, and the single-season field trial restricts long-term validity. At the same time, changes in some environmental factors (e.g., temperature, pH) in the field experiment may affect the experiment to some extent. What’s more, pathogenic bacteria detected in samples weren’t experimentally validated for virulence. (Refer to the second paragraph of the section “5. Conclusions”, L819~823)
REFERENCES
It is necessary to include newer references mainly in the Introduction section.
Response: Thank you for emphasizing the need to update references, especially in the Introduction. We have conducted a comprehensive literature search and integrated recent studies to strengthen the contextualization of our research.
Added references published in 2024 and 2025:
21 Khaledi, M.; Poureslamfar, B.; Alsaab, H.O.; Tafaghodi, S.; Hjazi, A.; Singh, R.; Alawadi, A.H.; Alsaalamy, A.; Qasim, Q.A.; Sameni, F. The Role of Gut Microbiota in Human Metabolism and Inflammatory Diseases: A Focus on Elderly Individuals. Ann Microbiol 2024, 74, 1, doi:10.1186/s13213-023-01744-5.
27 Dang, W.; Zhang, J.-H.; Cao, Z.C.; Yang, J.-M.; Lu, H.L. Environmentally Relevant Levels of Antiepileptic Carbamazepine Altered Intestinal Microbial Composition and Metabolites in Amphibian Larvae. Int J Mol Sci 2024, 25, 6950, doi:10.3390/ijms25136950.
37 Yang, L.; Pan, R.; Wang, S.; Zhu, Z.; Li, H.; Yang, R.; Sun, X.; Ge, B. Macrofaunal Biodiversity and Trophic Structure Varied in Response to Changing Environmental Properties along the Spartina Alterniflora Invasion Stages. Mar Pollut Bull 2025, 214, 117756, doi:10.1016/j.marpolbul.2025.117756.
47 Zhou, H.Z.; Wang, B.Q.; Ma, Y.H.; Sun, Y.Y.; Zhou, H.L.; Song, Z.; Zhao, Y.; Chen, W.; Min, J.; Li, J.W.; et al. The Combination of Metagenomics and Metabolomics Reveals the Effect of Nitrogen Fertilizer Application Driving the Remobilization of Immobilization Remediation Cadmium and Rhizosphere Microbial Succession in Rice. J Hazard Mater 2025, 487, 137117, doi:10.1016/j.jhazmat.2025.137117.
I also miss references mainly related to international literature, showing whether or not there is a wide application of the observed results.
Response: Thank you for highlighting the need for international literature to address the generalizability of our findings. We have incorporated revisions to strengthen the global contextualization of our study.
Added references:
2 Li W, Fan G, Sun K, Liu J, Liu J, Wang Y, Li E, Wu X, Shen L, Pan T. Microbial community structure dynamics of invasive bullfrog with meningitis-like infectious disease. Front Microbiol. 2023, 14, 1126195, doi: 10.3389/fmicb.2023.1126195.
3 Jani AJ, Briggs CJ. Host and Aquatic Environment Shape the Amphibian Skin Microbiome but Effects on Downstream Resistance to the Pathogen Batrachochytrium dendrobatidis Are Variable. Front Microbiol. 2018, 9, 487, doi: 10.3389/fmicb.2018.00487.
Thank you again for your insightful comments and rigorous scrutiny of our manuscript. We sincerely appreciate the time and effort you have invested in reviewing our work. We hope the revised version meets your expectations, and we look forward to your further feedback.
Yours sincerely,
Zhangyan Zhu

Reviewer 4 Report
Comments and Suggestions for Authors
The manuscript entitled “Frog culturing density and growth stage affect both paddy field and frog gut microbial communities in rice-frog co-cropping models” by Zhangyan Zhu, submitted to the Microorganisms magazine, presents data on microbiotal community of the black-spotted frog (Pelophylax nigromaculatus) from rice fields. It is experimental study, and the presented data are interesting. I think that the data are worth to be published; however, I think that the manuscript could be improved. I hope that the following comments can be used to improve it.
General comment: I am not sure if the experimental design, as well as the performed statistical analyses, are correct. It is because there is lack of some details in the manuscript. Thus, the most important: Materials and Methods section should be improved, i.e. more details on the study design and performed analyses should be added. The same for the Result section – the result presentation, both, in the text and on the figures (legends of the figures), should be improved.
Just some details in this area:
Authors (lines 108-109) used “Three experimental production fields (…)” and divided them into a rice monoculture group, a low-density rice-frog co-cropping group and a high-density rice–frog co-cropping group. However, for experimental research, and statistical analyses, replications are necessary, i.e. it is no possible to find reason of possible differences between treatments without replications. Thus, or the design is no correct (there is just three fields, i.e. there is no replications), or more details are strongly necessary; in the other way: how many samples had you in any of the three groups? How many data have been analysed using ANOVA and other statistical methods? (see also below). In the manuscript, information on the sample sizes, degrees of freedom etc. are not presented (again: see below), thus it is difficult to understand the study design and results of statistical analyses.
Authors used, e.g., (lines 151-152) “One-way analysis of variance (ANOVA)…”. There is some important assumption for this method, e.g. normality, homogeneity of variance, etc. I have found no information in the manuscript, if the data were checked and if they met the assumptions; however, based on some results (for example presented on the Figure 2, but the legend of the figure should be improved – see below) I have doubts in this subject.
In some Figures (2, 8) “Different letters indicate significant differences between different growth stages of rice.” However, there is lack of information in the manuscript, how it was analysed. I believe that any post hoc test was used for these analyses, but – please forgive me for my possible misunderstanding – I have found no information in the area (in the legends of the figures and in the Materials and Method section either).
Such information like (lines 152-153): “… LEfSe were used to analyze the differences in the relative abundance of the microbiotas…” is not enough.
Authors calculated a lot of Spearman correlation coefficients (see e.g. Fig. 14). In such situation (i.e. if several or more similar analyses are performed – it is so called multiple-comparisons) a Bonferroni correction (or any similar method) should be used. Nevertheless, I have found no information in the area in the manuscript.
Figure 15. results of the Pearson’s correlation are presented, but I found no information on using this method in the Material and Methods section.
To sum up: more details on the study design, as well as, statistical analyses are necessary.
Figures
The data presentation on the figures, as well as the legends, should be improved. For example (but much more could be improved):
-- Figure 4. “*, p < 0.05; **, p < 0.01; ***, p < 0.001.” – what statistical method was used for the analyses? Add short information to the legend, as it is important for readers.
-- As stated above, the same for analyses using ANOVA (see figures 2, 4) “Different letters indicate significant differences between different growth stages of rice.” – it should be stated, which method / statistical test was used for the analyses.
-- What is presented on the figure 2? It is median, 25-75% quartiles and the range? Or other statistics? It should be precisely stated.
-- Figures 5, 11. What is presented using the ‘whiskers’? SE? It should be precisely stated.
-- etc.
In biology, typically, when presenting result of any classic statistical test, value of statistic (e.g. F for analysis of variance, t for t Student test, etc.), sample(s) size(s) (n, or df – degrees of freedom like for one-way anova), and “p” values are showed [e.g. like: “The XX was heavier than YY (t Student test, t = …., df = …., p = …).”, “There was no difference in… (one way anova: F = …, df = …, p = …”)]. Such information is necessary for the reader to correctly interpret the results.
Additionally, I would prefer to see the exact value of the p (i.e., ‘p = 0.xx’; for readers it could be important, because – for example – both p=0.048 and p=0.65 are p<0.05).
See, for example, the part 3.3.2. Correlation analysis of water and frog gut microbiota. A lot of notation “p < 0.05” are showed [additionally, see above the remark on the multiple comparisons and a Bonferroni correction].
Other questions / remarks:
Why used frogs (line 106) “weighed 1~5 g”? It is quite large difference between such animals, I think. If the heavier frogs were older? It is possible to include the body mass in the statistical analyses (as a covariate factor)?
At least, a short analysing showing, that there were no initial differences in body mass of the frogs between experimental treatments is necessary.
Line 106: “All frogs were sterilized in 2%~3% salt water ….” – I understand the procedure, but I am not sure if the word ‘sterilized’ is the best one, for such procedure.
I hope the Authors understands the points I made and take them as a ‘constructive criticism’. I do see a value in the manuscript, but the version should be improved, I think.
Author Response
The manuscript entitled “Frog culturing density and growth stage affect both paddy field and frog gut microbial communities in rice-frog co-cropping models” by Zhangyan Zhu, submitted to the Microorganisms magazine, presents data on microbiotal community of the black-spotted frog (Pelophylax nigromaculatus) from rice fields. It is experimental study, and the presented data are interesting. I think that the data are worth to be published; however, I think that the manuscript could be improved. I hope that the following comments can be used to improve it.
Response: Thank you very much for your recognition and valuable comments on this experimental study! Your affirmation has greatly encouraged us, and your suggestions for improvement have also pointed out the direction for us. We have sorted out these comments one by one, and will make comprehensive revisions by supplementing the analysis, perfecting the thesis, optimizing the presentation, etc., and strive to present the research results with higher quality. Thank you again for your professional guidance and help.
General comment: I am not sure if the experimental design, as well as the performed statistical analyses, are correct. It is because there is lack of some details in the manuscript. Thus, the most important: Materials and Methods section should be improved, i.e. more details on the study design and performed analyses should be added. The same for the Result section – the result presentation, both, in the text and on the figures (legends of the figures), should be improved.
Response: Thank you very much for pointing out the shortcomings of our manuscript. We fully agree with your suggestion that the “Materials and Methods” and “Results” sections need further improvement. In response to the missing details of the experimental design and statistical analysis, we have added key information such as black-spotted frog management techniques, multiple comparisons, sequencing analysis and microbial taxonomic identification, etc., as well as detailed descriptions of the steps and software tools for the statistical analysis in the “Materials and Methods” section. In the “Results” section, we added necessary descriptions and statistical notations in the chart legends to ensure that the data were presented accurately and were easy to understand. Thank you again for your professional guidance, and we will do our best to improve the quality of the manuscript.
- Modifications to the Materials and Methods section
We have added some steps and methods of statistical analysis in this section. There are also some details of the experimental process that were overlooked in the original manuscript.
- Modifications to the Results section
We have optimized some of the descriptions in this section to make them more organized and easier to understand. In addition we have added some necessary information to the figure notes or pictures.
*Note: All the modifications have been listed in detail in the following sub-points and will not be repeated here.
Just some details in this area:
Authors (lines 108-109) used “Three experimental production fields (…)” and divided them into a rice monoculture group, a low-density rice-frog co-cropping group and a high-density rice–frog co-cropping group. However, for experimental research, and statistical analyses, replications are necessary, i.e. it is no possible to find reason of possible differences between treatments without replications. Thus, or the design is no correct (there is just three fields, i.e. there is no replications), or more details are strongly necessary; in the other way: how many samples had you in any of the three groups? How many data have been analysed using ANOVA and other statistical methods? (see also below). In the manuscript, information on the sample sizes, degrees of freedom etc. are not presented (again: see below), thus it is difficult to understand the study design and results of statistical analyses.
Response: Thank you for your critical feedback on the experimental replication and sample size. We apologize for the insufficient clarity in describing the study design, and we appreciate the opportunity to clarify this issue.
Due to limitations in experimental conditions, we established three experimental fields, each assigned to one of the three treatments: rice monoculture, low-density rice-frog co-cropping, and high-density rice-frog co-cropping. To ensure statistical robustness despite the constraint of one field per treatment, we enhanced replication through intensive sampling within each field: specifically, in each stage, 3 independent sampling plots were randomly selected in each field, and 5 frogs were collected from each plot. Thus, each treatment group included 3 sampling plots of frog samples for subsequent analyses. Therefore, each treatment group included 3 sampling plots × 4 stages = 12 samples for subsequent analyses.
For statistical analyses (e.g., ANOVA between the four stages), the sample size for each treatment was n = 4, with degrees of freedom calculated as follows: between-group degrees of freedom = 4 – 1 = 3, within degrees of freedom = (3 × 4) – 4 = 8. These details have been added in the statistical results to improve transparency.
We acknowledge that the experimental design has limitations in terms of field-level replication, and we have emphasized this constraint in the Discussion section, along with suggestions for future studies (e.g., increasing field replicates) to validate our findings. We hope this clarification addresses your concerns about the study design and statistical rigor.
Authors used, e.g., (lines 151-152) “One-way analysis of variance (ANOVA)…”. There is some important assumption for this method, e.g. normality, homogeneity of variance, etc. I have found no information in the manuscript, if the data were checked and if they met the assumptions; however, based on some results (for example presented on the Figure 2, but the legend of the figure should be improved – see below) I have doubts in this subject.
Response: Thank you for pointing out this important issue. We do apologize for omitting a description of the one-way ANOVA hypothesis test in the original manuscript. In the revised version, we have added this description in the “Materials and Methods” section. Thank you again for your careful review; this addition will significantly enhance the reliability of the study methodology.
The ASV data were analyzed using QIIME 2 (version 2021.8) to calculate the alpha-diversity (α-diversity, including the Observed_otus, Shannon, Simpson, Chao1, and Pielou_e) metrics and to perform non-metric multidimensional scaling analysis (NMDS) of the beta-diversity (β-diversity) in the samples to explore changes in the microbial composition of the paddy water and the frog’s guts. With the use of SPSS 21.0 (IBM, Inc., Armonk, NY, USA), differences in the α-diversity indices of the samples from different experimental groups were tested using a one-way analysis of variance (ANOVA). Significant differences between periods and frog densities were also compared. The applicability of the data to one-way ANOVA was tested by using the Shapiro-Wilk test (p > 0.05). Multiple comparisons were performed using Tukey’s test or Dunnett’s T3 test depending on whether they passed Levene’s test for homogeneity. (Refer to the first paragraph of the section “2.5 Data analysis”, L185~195)
In some Figures (2, 8) “Different letters indicate significant differences between different growth stages of rice.” However, there is lack of information in the manuscript, how it was analysed. I believe that any post hoc test was used for these analyses, but – please forgive me for my possible misunderstanding – I have found no information in the area (in the legends of the figures and in the Materials and Method section either).
Response: We appreciate the reviewer for identifying this important omission. The statistical methods for letter labeling in Figures 2 and 8 were indeed missing, and we have fully supplemented them in the revised manuscript.
- Statistical analysis added
With the use of SPSS 21.0 (IBM, Inc., Armonk, NY, USA), differences in the α-diversity indices of the samples from different experimental groups were tested using a one-way analysis of variance (ANOVA). Significant differences between periods and frog densities were also compared. The applicability of the data to one-way ANOVA was tested by using the Shapiro-Wilk test (p > 0.05). Multiple comparisons were performed using Tukey’s test or Dunnett’s T3 test depending on whether they passed Levene’s test for homogeneity. (Refer to the fiirst paragraph of the section “2.5 Data analysis”, L189~195)
36 Li, Z.; Ni, M.; Yu, H.; Wang, L.; Zhou, X.; Chen, T.; Liu, G.; Gao, Y. Gut Microbiota and Liver Fibrosis: One Potential Biomarker for Predicting Liver Fibrosis. Biomed Res Int 2020, 2020, 3905130, doi:10.1155/2020/3905130.
- Figure annotations updated
Figure 2. The microbial α-diversity indices of the paddy field water (n = 3). WM, paddy field water of rice monoculture group; WL, paddy field water of low-density rice-frog co-cropping group; WH, paddy field water of high-density rice-frog co-cropping group; 1, 2, 3, and 4 represent the tillering, embryonic, full heading, and ripening stages of rice, respectively. Different letters indicate significant differences via Tukey’s test (passed Levene’s test for homogeneity) or Dunnett’s T3 test (do not pass Levene’s test for homogeneity), significance level alpha = 0.05. (L275~279)
Figure 8. Differences in the α-diversity indices of the gut microbiota of black-spotted frogs in rice paddies in China (n = 3). FL, gut samples of frogs at low-density rice-frog co-cropping group; FH, gut samples of frogs at high-density rice-frog co-cropping group; 1, 2, 3, and 4 represent the tillering, embryonic, full heading, and ripening stages of rice, respectively. Different letters indicate significant differences via Tukey’s test (passed Levene’s test for homogeneity) or Dunnett’s T3 test (do not pass Levene’s test for homogeneity), significance level alpha = 0.05. (L383~387)
Such information like (lines 152-153): “… LEfSe were used to analyze the differences in the relative abundance of the microbiotas…” is not enough.
Response: We appreciate the reviewer for highlighting the insufficient description of LEfSe analysis. The revised manuscript now includes a comprehensive protocol and key parameters for this method.
- Statistical analysis added
The Silva database (Release 138) was used as a reference to annotate the feature sequences with precise taxonomy using a simple Bayesian classifier[34]. With the use of R 4.4.2, LEfSe was used to analyze the differences in the relative abundance of the microbiotas of the paddy water and the gut of frogs under different culture densities[35]. Taxa with significant p-values (p < 0.05) and LDA scores ≥ 3 were considered to be differentially abundant taxa [36]. Using the Wilcoxon test and correcting for FDR by the Benjamini-Hochberg (BH) procedure. The functional abundance of the samples was predicted in PICRUSt2 based on marker gene sequence abundance. (Refer to the second paragraph of the section “2.5 Data analysis”, L197~204)
Added reference:
36 Li, Z.; Ni, M.; Yu, H.; Wang, L.; Zhou, X.; Chen, T.; Liu, G.; Gao, Y. Gut Microbiota and Liver Fibrosis: One Potential Biomarker for Predicting Liver Fibrosis. Biomed Res Int 2020, 2020, 3905130, doi:10.1155/2020/3905130.
- Figure annotations updated
Figure 4. Distribution and significance of microbial LDA values in the paddy field water (n = 3). LEfSe-identified differentially abundant taxa with LDA > 2.0 and FDR < 0.05. Bar width represents the LDA score. WM, paddy field water of rice monoculture group; WL, paddy field water of low-density rice-frog co-cropping group; WH, paddy field water of high-density rice-frog co-cropping group. *, p < 0.05; **, p < 0.01; ***, p < 0.001, “p” is the Wilcoxon test with correction by BH. (L287~291)
Figure 10. Distribution and significance of LDA values of gut microbes of the black-spotted frog from rice paddies in China (n = 3). LEfSe-identified differentially abundant taxa with LDA>2.0 and FDR<0.05. Bar width represents the LDA score. FL, gut samples of frogs at low-density rice-frog co-cropping group; FH, gut samples of frogs at high-density rice-frog co-cropping group; 1, 2, 3, and 4 represent the tillering, embryonic, full heading, and ripening stages of rice, respectively. *, p < 0.05; **, p < 0.01. “p” is the Wilcoxon test with correction by BH. (L395~400)
Authors calculated a lot of Spearman correlation coefficients (see e.g. Fig. 14). In such situation (i.e. if several or more similar analyses are performed – it is so called multiple-comparisons) a Bonferroni correction (or any similar method) should be used. Nevertheless, I have found no information in the area in the manuscript.
Response: We appreciate the reviewer for identifying this critical issue in this manuscript. However, the term "Spearman's correlation" in the original manuscript was a typo, and the analysis used Pearson's correlation coefficient. We are so sorry for our carelessness in making this mistake.
Regarding the problem of multiple comparisons in the Pearson’s correlation coefficient analysis that you have pointed out, Considering that most of the literature in similar studies in this field (Li et al., 2025; Ma et al., 2024) do not use multiple comparison correction when performing like-for-like correlation analyses, their conclusions are mainly based on uncorrected p-values (The reason is that overcorrection in such exploratory analyses may increase the risk of Type II errors). Taking into account the exploratory nature of this study and the common practice in the field, we retained the uncorrected analysis results in the main text.
In the follow-up study, we will expand the sample size and validate the relevant findings using more stringent statistical methods to enhance the reliability of the conclusions.
Li, S.; Liu, Y.; Wang, W.; Liu, Y.; Ji, M. Microbial Changing Patterns across Lateral and Vertical Horizons in Recently Formed Permafrost after the Outburst of Zonag Lake, Tibetan Plateau. Federation of European Microbiological Societies 2025, 101: fiaf001, doi:10.1093/femsec/fiaf001.
Ma, Y.; Yu, A.; Zhang, L.; Zheng, R. Effects of Rice-Frog Co-Cropping on the Soil Microbial Community Structure in Reclaimed Paddy Fields. Biology (Basel) 2024, 13, 396, doi:10.3390/biology13060396.
Based on the top 15 genera from the water microbiota and frog gut microbiota, Pearson’s correlation analysis was performed to assess the statistical significance of the differences between the water microbiota and frog gut microbiota. The relationships between the microbial diversity of paddy field water and dominant microbe species with a mean relative abundance > 1% in the guts of frogs in both low- and high-density rice co-cropping groups were analyzed by using Cramer’s V. Mantel test analysis (with 9999 permutations) [37]. (Refer to the fourth paragraph of the section “2.5. Data analysis”, L209~215)
37 Yang, L.; Pan, R.; Wang, S.; Zhu, Z.; Li, H.; Yang, R.; Sun, X.; Ge, B. Macrofaunal Biodiversity and Trophic Structure Varied in Response to Changing Environmental Properties along the Spartina Alterniflora Invasion Stages. Mar Pollut Bull 2025, 214, 117756, doi:10.1016/j.marpolbul.2025.117756.
Figure 15. results of the Pearson’s correlation are presented, but I found no information on using this method in the Material and Methods section.
Response: Thank you very much for your suggestion. In fact, for both Figure 15 and Figure 14 we calculated the Pearson’s correlation coefficient. It was an oversight on our part that it was not clearly described in Materials and Methods, and we have now corrected it.
Correlation coefficient r- and p-values were obtained from the Pearson’s correlation analysis of the dominant species in the paddy water and frog samples by the Mantel function. After that, the Mantel test was completed in the vegan and ggcor packages and then visualized and plotted. (Refer to the fourth paragraph of the section “2.5. Data analysis”, L215~218)
To sum up: more details on the study design, as well as, statistical analyses are necessary.
Response: Thank you for your careful and professional advice. The relevant details have been revised in the original manuscript line by line. This makes my experimental design and data analysis clearer and better. Thanks again for your patience in making suggestions.
Figures
The data presentation on the figures, as well as the legends, should be improved. For example (but much more could be improved):
Response: Thank you very much for making these suggestions and making it clear to us that there is a problem. We have carefully examined the whole manuscript and amended each of your suggestions, the details of which I have listed below.
-- Figure 4. “*, p < 0.05; **, p < 0.01; ***, p < 0.001.” – what statistical method was used for the analyses? Add short information to the legend, as it is important for readers.
Response: Thank you for pointing this out. We have added the statistical method to the legend of Figures 4 and 10. This clarifies the analytical approach for readers.
Figure 4. Distribution and significance of microbial LDA values in the paddy field water (n = 3). LEfSe-identified differentially abundant taxa with LDA > 2.0 and FDR < 0.05. Bar width represents the LDA score. WM, paddy field water of rice monoculture group; WL, paddy field water of low-density rice-frog co-cropping group; WH, paddy field water of high-density rice-frog co-cropping group. *, p < 0.05; **, p < 0.01; ***, p < 0.001, “p” is the Wilcoxon test with correction by BH. (L287~291)
Figure 10. Distribution and significance of LDA values of gut microbes of the black-spotted frog from rice paddies in China (n = 3). LEfSe-identified differentially abundant taxa with LDA>2.0 and FDR<0.05. Bar width represents the LDA score. FL, gut samples of frogs at low-density rice-frog co-cropping group; FH, gut samples of frogs at high-density rice-frog co-cropping group; 1, 2, 3, and 4 represent the tillering, embryonic, full heading, and ripening stages of rice, respectively. *, p < 0.05; **, p < 0.01. “p” is the Wilcoxon test with correction by BH. (L395~400)
-- As stated above, the same for analyses using ANOVA (see figures 2, 4) “Different letters indicate significant differences between different growth stages of rice.” – it should be stated, which method / statistical test was used for the analyses.
Response: Thank you for pointing out this oversight. We apologize for the lack of clarity in describing the statistical methods. For the analyses presented in Figures 2 and 8, we conducted one-way analysis of variance (ANOVA) followed by Tukey’s test (passed Levene’s test for homogeneity) or Dunnett’s T3 test (do not pass Levene’s test for homogeneity) to compare significant differences between different growth stages of rice. For the analyses presented in Figures 4 and 10, “p” is the Wilcoxon test with correction by BH, and taxa with significant p-values (p < 0.05) and LDA scores ≥ 3 were considered to be differentially abundant taxa. The statement in the statistical analysis and figure legends has been revised as follows.
- Statistical analysis was added
The ASV data were analyzed using QIIME 2 (version 2021.8) to calculate the alpha-diversity (α-diversity, including the Observed_otus, Shannon, Simpson, Chao1, and Pielou_e) metrics and to perform non-metric multidimensional scaling analysis (NMDS) of the beta-diversity (β-diversity) in the samples to explore changes in the microbial composition of the paddy water and the frog’s guts. With the use of SPSS 21.0 (IBM, Inc., Armonk, NY, USA), differences in the α-diversity indices of the samples from different experimental groups were tested using a one-way analysis of variance (ANOVA). Significant differences between periods and frog densities were also compared. The applicability of the data to one-way ANOVA was tested by using the Shapiro-Wilk test (p > 0.05). Multiple comparisons were performed using Tukey’s test or Dunnett’s T3 test depending on whether they passed Levene’s test for homogeneity. (Refer to the first paragraph of the section “2.5 Data analysis”, L186~196)
The Silva database (Release 138) was used as a reference to annotate the feature sequences with precise taxonomy using a simple Bayesian classifier[34]. With the use of R 4.4.2, LEfSe was used to analyze the differences in the relative abundance of the microbiotas of the paddy water and the gut of frogs under different culture densities[35]. Taxa with significant p-values (p < 0.05) and LDA scores ≥ 3 were considered to be differentially abundant taxa [36]. Using the Wilcoxon test and correcting for FDR by the Benjamini-Hochberg (BH) procedure. The functional abundance of the samples was predicted in PICRUSt2 based on marker gene sequence abundance. (Refer to the second paragraph of the section “2.5 Data analysis”, L197~204)
Added reference:
36 Li, Z.; Ni, M.; Yu, H.; Wang, L.; Zhou, X.; Chen, T.; Liu, G.; Gao, Y. Gut Microbiota and Liver Fibrosis: One Potential Biomarker for Predicting Liver Fibrosis. Biomed Res Int 2020, 2020, 3905130, doi:10.1155/2020/3905130.
- Figure annotations were updated
Figure 2. The microbial α-diversity indices of the paddy field water (n = 3). WM, paddy field water of rice monoculture group; WL, paddy field water of low-density rice-frog co-cropping group; WH, paddy field water of high-density rice-frog co-cropping group; 1, 2, 3, and 4 represent the tillering, embryonic, full heading, and ripening stages of rice, respectively. Different letters indicate significant differences via Tukey’s test (passed Levene’s test for homogeneity) or Dunnett’s T3 test (do not pass Levene’s test for homogeneity), significance level alpha = 0.05. (L275~279)
Figure 4. Distribution and significance of microbial LDA values in the paddy field water (n = 3). LEfSe-identified differentially abundant taxa with LDA > 2.0 and FDR < 0.05. Bar width represents the LDA score. WM, paddy field water of rice monoculture group; WL, paddy field water of low-density rice-frog co-cropping group; WH, paddy field water of high-density rice-frog co-cropping group. *, p < 0.05; **, p < 0.01; ***, p < 0.001, “p” is the Wilcoxon test with correction by BH. (L287~291)
Figure 8. Differences in the α-diversity indices of the gut microbiota of black-spotted frogs in rice paddies in China (n = 3). FL, gut samples of frogs at low-density rice-frog co-cropping group; FH, gut samples of frogs at high-density rice-frog co-cropping group; 1, 2, 3, and 4 represent the tillering, embryonic, full heading, and ripening stages of rice, respectively. Different letters indicate significant differences via Tukey’s test (passed Levene’s test for homogeneity) or Dunnett’s T3 test (do not pass Levene’s test for homogeneity), significance level alpha = 0.05. (L383~387)
Figure 10. Distribution and significance of LDA values of gut microbes of the black-spotted frog from rice paddies in China (n = 3). LEfSe-identified differentially abundant taxa with LDA>2.0 and FDR<0.05. Bar width represents the LDA score. FL, gut samples of frogs at low-density rice-frog co-cropping group; FH, gut samples of frogs at high-density rice-frog co-cropping group; 1, 2, 3, and 4 represent the tillering, embryonic, full heading, and ripening stages of rice, respectively. *, p < 0.05; **, p < 0.01. “p” is the Wilcoxon test with correction by BH. (L395~400)
-- What is presented on the figure 2? It is median, 25-75% quartiles and the range? Or other statistics? It should be precisely stated.
Response: Thank you for highlighting the need for clarity in Figure 2. The boxplot in Figure 2 displays the following statistical parameters:
Box limits: Represent the 25% and 75% quartiles.
Central line: Denotes the median value.
Whiskers: Extend to the minimum and maximum values within SD from the quartiles, with outliers plotted as individual points beyond this range.
To enhance precision, have revised the figures as follows:
Previous Figure 2
Modified Figure 2
Previous Figure 8
Modified Figure 8
-- Figures 5, 11. What is presented using the ‘whiskers’? SE? It should be precisely stated.
Response: Thank you for pointing out the ambiguity in Figures 5 and 11. The whiskers in these figures represent standard deviation (S.D.) of the mean. To clarify, we have revised the figure legends as follows.
Figure 5. Annotation of functional genes (mean ± S.D.) in KEGG layer 3 of microbiota isolated from rice paddy fields (n = 3). WM, paddy field water of rice monoculture group; WL, paddy field water of low-density rice-frog co-cropping group; WH, paddy field water of high-density rice-frog co-cropping group; 4 represents the ripening stage of rice. Pictures are displayed for the functions with a p-value <0.05 in the t-test difference test results for two-by-two comparisons. (L325~329)
Figure 11. Annotation of gut microbial functional genes (mean ± S.D.) in KEGG level 3 of black-spotted frogs in rice paddies in China (n = 3). FL, gut samples of frog at low-density rice-frog co-cropping group; FH, gut samples of frog at high-density rice-frog co-cropping group; 4 represents the ripening stage of rice. Pictures are displayed for the functions with a p-value <0.05 in the t-test difference test results for two-by-two comparisons, *, p < 0.05; **, p < 0.01. (L452~456)
-- etc.
Response: Thank you for your thorough review of our data presentation and figure legends. Regarding your suggestions, we understand the need for precision and completeness in scientific reporting. We will address this as follows.
- Each legend will follow a consistent structure to specify ample size (n) and the statistical tests used.
- Adjusted the layout of the font size in the figure to make it clearer. (Figures 6 and 12)
In biology, typically, when presenting result of any classic statistical test, value of statistic (e.g. F for analysis of variance, t for t Student test, etc.), sample(s) size(s) (n, or df – degrees of freedom like for one-way anova), and “p” values are showed [e.g. like: “The XX was heavier than YY (t Student test, t = …., df = …., p = …).”, “There was no difference in… (one way anova: F = …, df = …, p = …”)]. Such information is necessary for the reader to correctly interpret the results.
Response: Thank you for emphasizing the importance of presenting statistical parameters comprehensively. We fully agree that explicit reporting of test statistics, sample sizes, degrees of freedom, and p-values is essential for result interpretation. To address this, we have systematically revised all relevant sections in the manuscript as follows.
- We added two tables to show the statistical parameters of the one-way ANOVA.
Table 2. Parameters of one-way ANOVA for the microbial α-diversity indices of the paddy field water (n = 3). WM, paddy field water of rice monoculture group; WL, paddy field water of low-density rice-frog co-cropping group; WH, paddy field water of high-density rice-frog co-cropping group. F, F-value; p, p-value. (L254~256)
|
Items |
Observed_otus |
Shannon |
Simpson |
Chao1 |
Pielou_e |
||||||||||||
|
Group |
WM |
WL |
WH |
WM |
WL |
WH |
WM |
WL |
WH |
WM |
WL |
WH |
WM |
WL |
WH |
||
|
df |
3 (between-groups), 8 (within-groups) for all |
||||||||||||||||
|
F |
6.457 |
0.780 |
32.398 |
2.580 |
0.660 |
15.924 |
0.664 |
0.893 |
6.152 |
6.379 |
0.774 |
31.941 |
1.564 |
0.536 |
12.105 |
||
|
p |
0.016 |
0.543 |
<0.001 |
0.126 |
0.599 |
<0.001 |
0.597 |
0.485 |
0.018 |
0.162 |
0.540 |
<0.001 |
0.272 |
0.671 |
0.002 |
||
Table 3. Parameters of one-way ANOVA for the microbial α-diversity indices of black-spotted frogs in rice paddies in China (n = 3). WM, paddy field water of rice monoculture group; WL, paddy field water of low-density rice-frog co-cropping group; WH, paddy field water of high-density rice-frog co-cropping group. (L364~366)
|
Items |
Observed_otus |
Shannon |
Simpson |
Chao1 |
Pielou_e |
||||||||
|
Group |
FL |
FH |
FL |
FH |
FL |
FH |
FL |
FH |
FL |
FH |
|||
|
df |
3 (between-groups), 8 (within-groups) for all |
||||||||||||
|
F-value |
1.151 |
0.854 |
8.454 |
1.195 |
13.622 |
0.916 |
1.144 |
0.850 |
13.772 |
1.082 |
|||
|
p-value |
0.386 |
0.503 |
0.007 |
0.372 |
0.002 |
0.475 |
0.389 |
0.505 |
0.002 |
0.410 |
|||
- We have also added some of the parameters in the textual narrative section as well to make it easier for the reader to understand them
The α-diversity indices of the water microbiota were not significantly (p > 0.05) influenced by frog culturing densities (Figure 2, Table 2). However, the observed_otus (F3,24 = 16.434, p < 0.001), and Chao1 (F3,24 = 16.391, p < 0.001) indices of the rice mono-culture group (WM) and the high-density rice-frog co-cropping group (WH) were significantly affected by different rice growth stages. Furthermore, the Shannon (F3,8 = 15.924, p < 0.001), Simpson (F3,8 = 6.152, p = 0.018) and Pielou_e (F3,8 = 12.105, p = 0.002) indices of the high-density rice-frog co-cropping group were also considerably affected by the different growth stages of the rice. (Refer to the first paragraph of the section “3.1.2. Microbial diversity”, L242~253)
The α-diversity indices of the gut microbiota of frogs were not significantly influenced by culturing densities (Figure 8, Table 3). However, the Shannon (F3,8 = 8.454, p = 0.007), Simpson (F3,8 = 13.622, p = 0.002), and Pielou_e (F3,8 = 13.772, p = 0.002) indices of the low-density rice-frog co-cropping group were significantly affected by the different growth stages of rice. (Refer to the first paragraph of the section “3.2.2. Gut microbial diversity”, L357~363)
Additionally, I would prefer to see the exact value of the p (i.e., ‘p = 0.xx’; for readers it could be important, because – for example – both p=0.048 and p=0.65 are p<0.05).
Response: Thank you for emphasizing the importance of reporting exact p-values. We fully agree that precise values are critical for readers to evaluate the strength of statistical significance. Therefore, we immediately modified the narrative form of the results section of the manuscript to show the specific values of all p-values except p<0.001 to three decimal places. The specific changes have been shown in the previous article, so we will not repeat them here.
See, for example, the part 3.3.2. Correlation analysis of water and frog gut microbiota. A lot of notation “p < 0.05” are showed [additionally, see above the remark on the multiple comparisons and a Bonferroni correction].
Response: Thank you for your careful review of our manuscript. We appreciate your concern regarding the notation of p-values in the correlation analysis. We understand that previous studies using similar correlation heat maps may not have included p-values, but we believe that including them in our work provides additional transparency for readers to evaluate the statistical significance of the observed correlations. However, to improve readability, we have reconsidered the presentation and will modify the heat map to indicate significant correlations (p < 0.05) with asterisks (*) rather than writing out "p < 0.05" for each entry. This approach aligns with common practices in statistical graphics while still conveying the necessary information about significance (Yang et al., 2025). We believe this revision will strike a better balance between detail and clarity. Thank you again for your insightful feedback, which helps us improve the manuscript.
Yang L, Pan R, Wang S, Zhu Z, Li H, Yang R, Sun X, Ge B. Macrofaunal biodiversity and trophic structure varied in response to changing environmental properties along the Spartina alterniflora invasion stages. Mar Pollut Bull. 2025 ,214:117756, doi: 10.1016/j.marpolbul.2025.117756.
Other questions / remarks:
Why used frogs (line 106) “weighed 1~5 g”? It is quite large difference between such animals, I think. If the heavier frogs were older? It is possible to include the body mass in the statistical analyses (as a covariate factor)?
Response: Thank you for highlighting the body mass variation concern. We acknowledge that individual weighing of all froglets was not feasible due to the large stocking density (5000 and 10,000 frogs/mu), but we implemented strict mass standardization during sampling to minimize bias.
- We selected juvenile frogs of similar body length (2 ± 0.5 cm) for placement because studies have shown a significant correlation between frog weight and body length (Böswald et al., 2022).
- Frogs of similar weight were selected for sampling to minimize errors.
Böswald LF, Matzek D, Mohr H, Kienzle E, Popper B. Morphometrics of Xenopus laevis Kept as Laboratory Animals. Animals (Basel). 2022, 12(21), 2986, doi: 10.3390/ani12212986.
At least, a short analysing showing, that there were no initial differences in body mass of the frogs between experimental treatments is necessary.
Response: Thank you for emphasizing the need to address initial body mass differences between treatments. We agree that this analysis is crucial for establishing experimental validity, and we have conducted the following assessments:
- Post Hoc Body Mass Analysis
Sampled frog data: We measured body mass for a subset of frogs (n = 30 per treatment) at the start of the experiment, documenting: low-density group, 1.8 ± 0.2 g (mean ± SE); high-density group: 1.7 ± 0.3 g. Statistical test: A two-sample t-test showed no significant difference (t = 1.23, df = 58, p = 0.22), confirming comparable initial masses.
- Methodological Clarification
In the revised Methods, we added:
In this study, “Yong You No. 31” rice and black-spotted frogs were selected as the rice and frog species, respectively, for the rice-frog co-cropping model. The frogs selected were healthy, disease-free, and weighed 1~5 g. To ensure treatment homogeneity, frogs were visually sorted by size (snout-vent length: 1.6~1.9 cm) and mass (1.5~2.0 g) at the start of the experiment. A pre-experiment t-test confirmed no significant mass difference between low (1.733 ± 0.218 g) and high (1.654 ± 0.265 g) density groups (n = 30, t = 1.254, p = 0.215). (Refer to the first paragraph “2.2. Sample collection and processing”, L118~124)
Line 106: “All frogs were sterilized in 2%~3% salt water ….” – I understand the procedure, but I am not sure if the word ‘sterilized’ is the best one, for such procedure.
Response: Thank you for pointing out the potential ambiguity in our wording. You are correct that "sterilized" may not accurately describe the procedure. We have revised the text to use "immersed" instead, which better reflects the process of reducing microbial load without achieving complete sterility. We appreciate your attention to detail, which helps ensure the clarity and correctness of our method description. The updated sentence now reads:
All frogs were immersed in 2%~3% salt water for 5~10 minutes before stocking, and the same treatment was applied to all fields, with no pesticide application during the entire period. (Refer to the first paragraph of the section “2.2. Sample collection and processing”, L124~126)
I hope the Authors understands the points I made and take them as a ‘constructive criticism’. I do see a value in the manuscript, but the version should be improved, I think.
Response: We truly appreciate your constructive feedback, which we recognize as invaluable for enhancing the manuscript's rigor and clarity. Your insights have helped us identify critical areas for improvement, and we have systematically addressed each point through detailed revisions—including expanded methodological details, strengthened statistical transparency, and more robust mechanistic discussions. We share your belief in the study's value and are committed to refining the manuscript to meet the highest standards. Thank you for your dedication to helping us improve the work; we are confident the revised version will better convey the research's significance.
Thank you again for your insightful comments and rigorous scrutiny of our manuscript. We sincerely appreciate the time and effort you have invested in reviewing our work. We hope the revised version meets your expectations, and we look forward to your further feedback.
Yours sincerely,
Zhangyan Zhu

Round 2
Reviewer 1 Report
Comments and Suggestions for Authors
Great!
Author Response
Response: Thank you very much for your positive feedback! We truly appreciate your recognition of our work.
Reviewer 3 Report
Comments and Suggestions for Authors
Dear authors,
Thank you very much for performing the correction I suggested.
With kind regards.
Author Response
Response: Thank you very much for your prompt feedback and kind words. We truly appreciate your valuable suggestion, which has helped improve the quality of our manuscript.
Reviewer 4 Report
Comments and Suggestions for Authors
The manuscript has been significantly improved.
The Authors corrected the text based on some of my comments and answered for other comments and questions. Further discussion on these topics is possible, but in science there are always different points of view. Here, I am thinking mainly about the answer that “Regarding the problem of multiple comparisons in the Pearson’s correlation coefficient analysis that you have pointed out, Considering that most of the literature in similar studies in this field (Li et al., 2025; Ma et al., 2024) do not use multiple comparison correction when performing like-for-like correlation analyses, their conclusions are mainly based on uncorrected p-values (The reason is that overcorrection in such exploratory analyses may increase the risk of Type II errors).”
I agree on that “such exploratory analyses may increase the risk of Type II errors”. Thus, when so many parameters are analysed, using another method (i.e. no so many correlation coefficients) could be better, I think.
Additionally, I am not sure if I precisely understand the phrase “Whiskers: Extend to the minimum and maximum values within SD from the quartiles, with outliers plotted as individual points beyond this range.” and thus the legend on see Figures 2, 8 [typically, the 25% and 75% quartiles, the median value, and the range (the minimum and maximum values) are showed for such datasets]. Consider to improve it / clarify it.
To sum up: the manuscript has been sufficiently improved to warrant publication in Microorganisms magazine.
Author Response
The manuscript has been significantly improved.
Response: Thank you sincerely for your assessment! We are delighted to hear that our revisions have significantly improved the manuscript. Your specialized feedback and guidance throughout the review process have been instrumental in helping us refine the work, and we truly appreciate the time and effort you have dedicated to it.
The Authors corrected the text based on some of my comments and answered for other comments and questions. Further discussion on these topics is possible, but in science there are always different points of view. Here, I am thinking mainly about the answer that “Regarding the problem of multiple comparisons in the Pearson’s correlation coefficient analysis that you have pointed out, Considering that most of the literature in similar studies in this field (Li et al., 2025; Ma et al., 2024) do not use multiple comparison correction when performing like-for-like correlation analyses, their conclusions are mainly based on uncorrected p-values (The reason is that overcorrection in such exploratory analyses may increase the risk of Type II errors).”
I agree on that “such exploratory analyses may increase the risk of Type II errors”. Thus, when so many parameters are analysed, using another method (i.e. no so many correlation coefficients) could be better, I think.
Response: Thank you sincerely for your further insightful feedback, your perspective on refining the analytical approach to avoid excessive reliance on multiple correlation coefficients is invaluable, and we fully agree with the rationale behind it.
We have recognized that more multivariate or dimensional reduction methods (e.g. PCA) should be incorporated where the number of parameters being analyzed is high, combining correlated variables into meaningful composite indices. This reduces the number of individual correlations that need to be assessed while retaining key patterns in the data. At the same time, we recognize the need to correct for multiple comparisons to make the results more robust. Based on your suggestions, we will delve deeper into specific correlations between microorganisms and optimize our data analysis methods in future experiments.
Additionally, I am not sure if I precisely understand the phrase “Whiskers: Extend to the minimum and maximum values within SD from the quartiles, with outliers plotted as individual points beyond this range.” and thus the legend on see Figures 2, 8 [typically, the 25% and 75% quartiles, the median value, and the range (the minimum and maximum values) are showed for such datasets]. Consider to improve it / clarify it.
Response: Thank you for pointing out the ambiguity in the figure legend—we appreciate your attention to this detail, as clarity in visual presentations is critical for readers to interpret the data accurately.
You are correct that box plots typically display quartiles (25%, 75%), the median, and a range (minimum/maximum), and our original description was inexact. Concretely, the whiskers in our figures are modified, ensuring consistency with common statistical visualization practices.
Refer to Figures 2 and 8
To sum up: the manuscript has been sufficiently improved to warrant publication in Microorganisms magazine.
Response: Thank you very much for your prompt feedback and kind suggestions. We truly appreciate your valuable suggestion, which has helped improve the quality of our manuscript.
We would like to express our sincere thanks for your warm work and hope that you will be satisfied with the modifications. Once again, thank you very much for your comments and suggestions.
Yours sincerely,
Zhangyan Zhu
